# HAT cofactor TRRAP modulates microtubule dynamics via SP1 signaling to prevent neurodegeneration

Alicia Tapias[1†], David Lázaro[1†], Bo-Kun Yin[1†], Seyed Mohammad Mahdi Rasa[1], Anna Krepelova[1], Erika Kelmer Sacramento[1], Paulius Grigaravicius[1], Philipp Koch[1], Joanna Kirkpatrick[1], Alessandro Ori[1], Francesco Neri[1], Zhao-Qi Wang[1,2]*

[1]Leibniz Institute on Aging – Fritz Lipmann Institute (FLI), Jena, Germany; [2]Faculty of Biological Sciences, Friedrich-Schiller-University of Jena, Jena, Germany

**Abstract** Brain homeostasis is regulated by the viability and functionality of neurons. HAT (histone acetyltransferase) and HDAC (histone deacetylase) inhibitors have been applied to treat neurological deficits in humans; yet, the epigenetic regulation in neurodegeneration remains elusive. Mutations of HAT cofactor TRRAP (transformation/transcription domain-associated protein) cause human neuropathies, including psychosis, intellectual disability, autism, and epilepsy, with unknown mechanism. Here we show that Trrap deletion in Purkinje neurons results in neurodegeneration of old mice. Integrated transcriptomics, epigenomics, and proteomics reveal that TRRAP via SP1 conducts a conserved transcriptomic program. TRRAP is required for SP1 binding at the promoter proximity of target genes, especially microtubule dynamics. The ectopic expression of Stathmin3/4 ameliorates defects of TRRAP-deficient neurons, indicating that the microtubule dynamics is particularly vulnerable to the action of SP1 activity. This study unravels a network linking three well-known, but up-to-date unconnected, signaling pathways, namely TRRAP, HAT, and SP1 with microtubule dynamics, in neuroprotection.

*For correspondence:
Zhao-Qi.Wang@leibniz-fli.de

[†]These authors contributed equally to this work

## Introduction

Neurodegenerative diseases are a range of incurable and debilitating conditions strongly linked with age, which represent a social and economic burden given the burgeoning elderly population. The key features of the brain are its adaptability and plasticity, which facilitate rapid, coordinated responses to changes in the environment, all of which require delicate brain functionality and maintenance. Progressive neuronal loss, synaptic deficits, disintegration of neuronal networks due to axonal and dendritic retraction, and failure of neurological functions are hallmarks of neurodegeneration (*Palop et al., 2006*; *Gan et al., 2018*). Molecular causes of neurodegeneration are believed to include protein misfolding and degradation, neuroinflammation, oxidative stress, DNA damage accumulation, mitochondrial dysfunction, as well as programmed cell death (*Palop et al., 2006*; *Gan et al., 2018*; *Kurtishi et al., 2019*; *Burté et al., 2015*; *Chi et al., 2018*).

Epigenetic mechanisms, including DNA methylation, histone modifications, non-coding or small RNAs, have been linked to brain development and neurological disorders, such as autism, intellectual disability (ID), and epilepsy, as well as neurodegenerative processes (*Meaney and Ferguson-Smith, 2010*; *Berson et al., 2018*; *Christopher et al., 2017*; *Tapias and Wang, 2017*). Histone acetylation, which is modulated by a range of histone acetyltransferase (HAT) families, is a major epigenetic modification controlling a wide range of cellular processes (*Tapias and Wang, 2017*; *Choudhary et al., 2014*). HATs and histone deacetylases (HDACs) maintain a proper acetylation kinetics of histones, yet also other protein substrates, which can coordinate histone dynamics over

large regions of chromatin to regulate the global gene expression as well as target gene-specific regions or promoters (*Vogelauer et al., 2000*; *Nagy and Tora, 2007*). HAT-HDAC-mediated histone modifications have been suggested to play a role in brain functionality, including memory formation, mood, drug addiction, and neuroprotection (*Meaney and Ferguson-Smith, 2010*; *Berson et al., 2018*; *Christopher et al., 2017*; *Delgado-Morales et al., 2017*; *Levenson and Sweatt, 2005*; *Renthal and Nestler, 2008*). For example, the pharmacological inhibition of HDACs has been used for their anti-epileptic, anti-convulsive, and mood-stabilizing effects (*Chiu et al., 2013*). In addition, HDAC inhibitors or HAT activators have been employed in clinics to treat the neurological symptoms of neurodegenerative diseases and psychiatric disorders, as well as autism, memory loss, and cognitive function, although not always successfully (*Christopher et al., 2017*; *Delgado-Morales et al., 2017*; *Selvi et al., 2010*; *Ganai et al., 2016*). Genome-wide approaches have revealed global and local changes in multiple histone marks; yet, the impact and meaning of these alterations in the pathophysiological processes are obscure. A major hurdle is the lack of specificity of these pharmacological interventions causing adverse side effects in clinical treatment. Interestingly, alterations of the acetylation profiles have been found in neurodegenerative disorders including Huntington's disease (HD) (reviewed in *Valor, 2017*), Amyotrophic lateral sclerosis (ALS) (reviewed in *Garbes et al., 2013*), spinal muscular atrophy (SMA) (*Kernochan et al., 2005*), Parkinson's disease (PD) (*Harrison et al., 2018*; *Park et al., 2016*), and Alzheimer's disease (AD) (*Klein et al., 2019*; *Marzi et al., 2018*). These findings highlight the involvement of HATs in the etiology of neurodegenerative processes, yet through various mechanisms (*Cobos et al., 2019*). Although widely discussed (*Saha and Pahan, 2006*; *Konsoula and Barile, 2012*), the role of histone acetylation and deacetylation in the adult central nervous system (CNS) and brain homeostasis remains largely unknown.

To gain insight into how the alteration of histone acetylation maintains neuronal homeostasis and prevents neurodegeneration, we used mouse and cellular models, in which the HAT essential cofactor TRRAP is deleted, so that the general HAT activity is disturbed. TRRAP (transformation/transcription domain-associated protein) interacts with E2F1 and c-Myc at gene promoters and is a critical component shared by several HAT complexes, including those from the GNAT and MYST families, which facilitates the recruitment of HAT complexes to target proteins for acetylation (*Tapias and Wang, 2017*; *Knutson and Hahn, 2011*). The complete deletion of Trrap in mice and cells is incompatible with the life of proliferating cells and mouse development, because of severe defects in the spindle checkpoint and cell cycle control (*Herceg et al., 2001*; *Dhanalakshmi et al., 2004*). A tissue specific deletion of Trrap in embryonic neural stem cells leads to a dysregulation of the cell cycle length which drives the premature differentiation of neuroprogenitors (*Tapias et al., 2014*). In humans, missense variants of TRRAP have been recently reported to associate with neuropathological symptoms, including psychosis, ID, autism spectrum disorder (ASD), and epilepsy (*Cogné et al., 2019*; *Mavros et al., 2018*). These basic and clinical studies point to a potential involvement of TRRAP in the manifestation of these neuropathies in humans. Because the known function of TRRAP in the mitotic checkpoint and cell cycle control does not apply to postmitotic cells, i.e., neurons, how TRRAP and its mediated HAT regulate adult neuronal fitness and affect neurodegeneration remains elusive.

In this study, we attempt to elucidate the molecular pathways that are governed by Trrap-HAT in postmitotic neural tissues. We find that Trrap deletion in the mouse model (*Mus musculus*) causes an age-dependent loss of existing neurons leading to neurodegeneration. We show that Trrap-HAT specifically regulates the Sp1 pathway that controls various neural processes, among which microtubule dynamics is particularly affected. Our study discloses the Trrap-HAT-Sp1 axis as a novel regulator of neuronal arborization and neuroprotection.

## Results

### *Trrap* deletion in Purkinje cells results in cerebellar degeneration

To study the role of histone acetylation and Trrap in postmitotic neurons, we generated two mouse models. First we crossed mice carrying the *Trrap* floxed allele (*Trrap*f/f) (*Herceg et al., 2001*) with *Pcp2-Cre* mice (Tg(Pcp2-cre)2Mpin) (*Barski et al., 2000*), to delete *Trrap* in Purkinje cells (Trrap-PCΔ). *Trrap*+/f mice with the Cre transgene, or *Trrap*f/f without the Cre transgene, were phenotypically normal, and thus they were used as controls. Trrap-PCΔ mice were born healthy and exhibited

normal (in rotarod tests) or mild defective (in beam balance) motor coordination at young age (1–2 months). However, they displayed an evident miscoordination at mid age (3–6 months), which became more severe after 9 months (old group) (*Figure 1A*). By the age of 1 year, Trrap-PCΔ mice developed an age-dependent locomotor dysfunction characterized by signs of ataxia, namely impaired coordination and unsteady gait (data not shown).

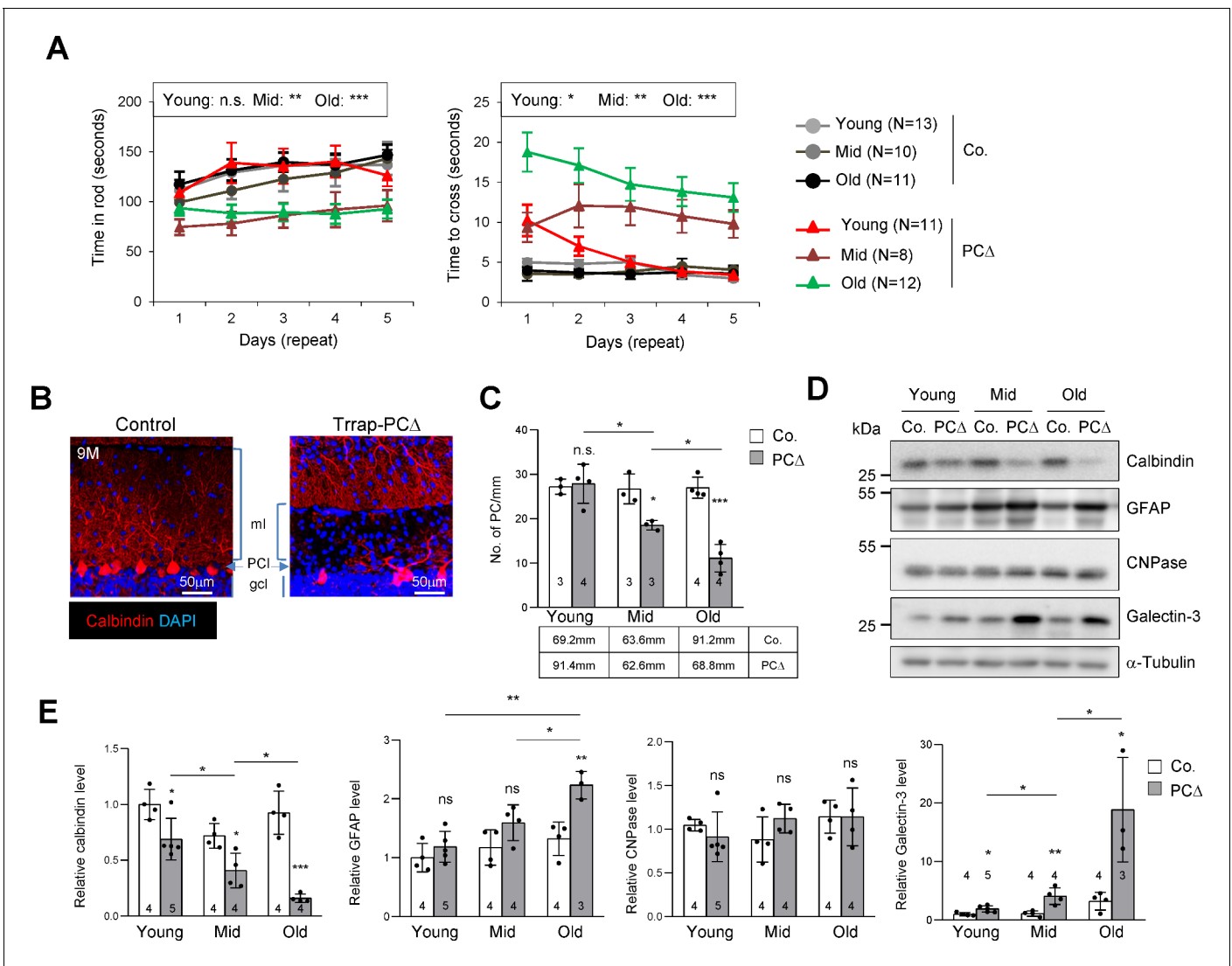

**Figure 1.** Deletion of Trrap in Purkinje cells causes neurodegeneration. (**A**) The rotarod test and the beam balance were used to assess the motor coordination of control and Trrap-PCΔ mice. The left panel depicts the time that the mice stayed in the rod before falling off. The right panel shows the quantification of the time taken by mice to cross the beam. (N) Indicates the number of mice analyzed. Young: 1–2 months; mid age: 3–6 months; old: 9–12 months. (**B**) Immunostaining of the cerebellar sections of 9-month-old mice using an antibody against calbindin (red, Purkinje cells). ml: molecular layer; gcl: granule cell layer; PCl: Purkinje Cell layer. (**C**) The quantification of the number of Purkinje cells in the cerebellum at the indicated ages. The mm of the Purkinje cell layer analyzed are indicated in the table. Young: 1–2 months; mid age: 3–6 months; old: 9–12 months. (**D**) The representative of the western blot analysis for markers for Purkinje cells (calbindin), astrocytes (GFAP), oligodendrocytes (CNPase), and activated microglia (Galectin-3). (**E**) The quantification of western blots of the cerebellum (from **D**). Signal intensities are normalized to α-Tubulin. The numbers inside the bars indicate the number of mice analyzed. Young: 1 month; mid age: 4 months; old: 9 months. Co.: control; PCΔ: Trrap-PCΔ. Mean ± standard error of the mean is shown. Two-way ANOVA and Holm–Sidak test was performed for statistical analysis in (**A**), (**B**) and Student's t-test or one-way ANOVA for (**C**), (**E**). n.s.: not significant. *p≤0.05, **p≤0.01, ***p≤0.001.

The online version of this article includes the following figure supplement(s) for figure 1:

**Figure supplement 1.** Deletion of Trrap in Purkinje cells causes neurodegeneration.

Immunostaining of the Purkinje cell marker calbindin revealed a progressive loss of Purkinje cells in Trrap-PCΔ cerebella – starting from 3 months old mice – which became severe at 9 months (*Figure 1B,C*). Western blot analysis of mutant brains of different ages confirmed a progressive Purkinje cell loss, as judged by the downregulation of calbindin (*Figure 1D,E*). While there was no significant change in the expression of the oligodendrocyte marker CNPase, a progressive increase of the astrocyte marker GFAP, namely astrogliosis, and the activated microglia marker Galectin-3 were evident in mutant cerebella, both of which are hallmarks of neurodegeneration (*Figure 1D,E*). Immunostaining of the cerebella of 2-month-old and 9-month-old mice confirmed a loss of Purkinje cells and a great increase of astrocytes (GFAP+ signals), a sign of astrogliosis, in Trrap-PCΔ cerebella at old age, whereas Trrap-PCΔ cerebella of young mice were normal (*Figure 1—figure supplement 1a,b*). Also, TUNEL staining detected more cell death in all cerebellar lobes of old mice (*Figure 1—figure supplement 1a,b*). The microglia activation and astrogliosis could be due to a homeostatic response to Purkinje cell loss, but they closely resemble neurodegeneration.

## *Trrap* deleted Purkinje cells exhibit age-dependent axonal swellings and dendrite retraction

To examine the neurodegenerative process, we analyzed *Trrap*-deleted Purkinje cells during early postnatal life. Immunostaining of brain sections using antibodies against calbindin and myelin-binding protein (MBP) detected axonal swellings of Trrap-deleted Purkinje cells readily at 1 month of age, prior to Purkinje cell loss (*Figure 2A*). Axonal swellings were generally myelinated at this age (*Figure 2A*), although a loss of myelination was observed occasionally in a few severe cases (data not shown). Furthermore, transmission electron microscopy (TEM) revealed the myelination index of Purkinje cell axons as normal in young (1-month-old) mice, but significantly lower, as judged by a higher g-ratio, in mid age (6-month-old) Trrap-PCΔ mice compared to controls (*Figure 2B,C*).

To quantify the morphological changes of *Trrap*-deleted Purkinje cells, we generated Trrap-PCΔ mice expressing a Confetti transgene (B6.Cg-Tg(Thy1-Brainbow1.0)HLich/J) (thereafter Trrap-PCΔ-Confetti), which enables individual Purkinje cells to stochastically express one of four fluorescent proteins upon Cre expression (*Snippert et al., 2010*). This allows distinguishing single neuron morphology from adjacent cells and reconstructing the dendritic trees of individual Purkinje cells (*Figure 2D*). A Sholl analysis (*Figure 2—figure supplement 1a*) of *Trrap*-deleted Purkinje cells at young age (1–4 months) and at old age (10 months) showed a progressive decrease in the size of their dendritic trees without great effects on their complexity as judged by the critical value (*Figure 2E,F*, *Figure 2—figure supplement 1b*). Consistent with the Sholl analysis data, the molecular layer became thinner at older age compared to young age (*Figure 2G*, *Figure 1—figure supplement 1b*). These observations indicate that Trrap deletion does not affect arborization, but rather causes a retraction of already formed dendrites of neurons.

## Trrap deletion changes transcriptional programs in neurons

The scarcity of Purkinje cell neurons in the Trrap-PCΔ cerebellar model limited the searching for the molecular mechanism related to the Trrap-HAT function. To gain a feasible approach, we devised another mouse model in which Trrap was deleted in a large subset of neurons, which would facilitate the molecular analysis of the HAT function in the brain. To this end, we crossed *Trrap*^f/f mice with *Camk2-Cre* transgenic mice (Tg(Camk2a-cre/ERT2)2Gsc) to generate mice with a Trrap deletion in pyramidal neurons in the cortex and striatum of the forebrain (designated as Trrap-FBΔ). Trrap-FBΔ brains were normal and had an efficient deletion of Trrap already at day 10 of the postnatal life (P10) (*Figure 3A*, see below for protein analysis in *Figure 4A,B*, and for qPCR analysis *Figure 4—figure supplement 1f*). We then carried out RNA-seq and proteomic analyses using cortices and striata from P10 Trrap-FBΔ and control mice. Trrap deletion resulted in highly reproducible changes in the transcriptome of cortices and striata with 5090 and 4389 differentially expressed genes (DEGs) respectively (cutoff adjusted, p<0.05) (*Figure 3B*, *Figure 3—figure supplement 1a*, *Supplementary file 1*). The Trrap-FBΔ cortex and striatum shared 2695 common DEGs, corresponding to 52.9% and 61.4% of the respective tissues. The directionality of the changes was conserved in 99.3% of the genes (*Figure 3C*, *Supplementary file 1*). Among the common DEGs, 1122 upregulated and 1554 downregulated genes were overlapping between these two parts of the brain (*Figure 3C*). These results strongly suggest that similar mechanisms operate in the neurons from

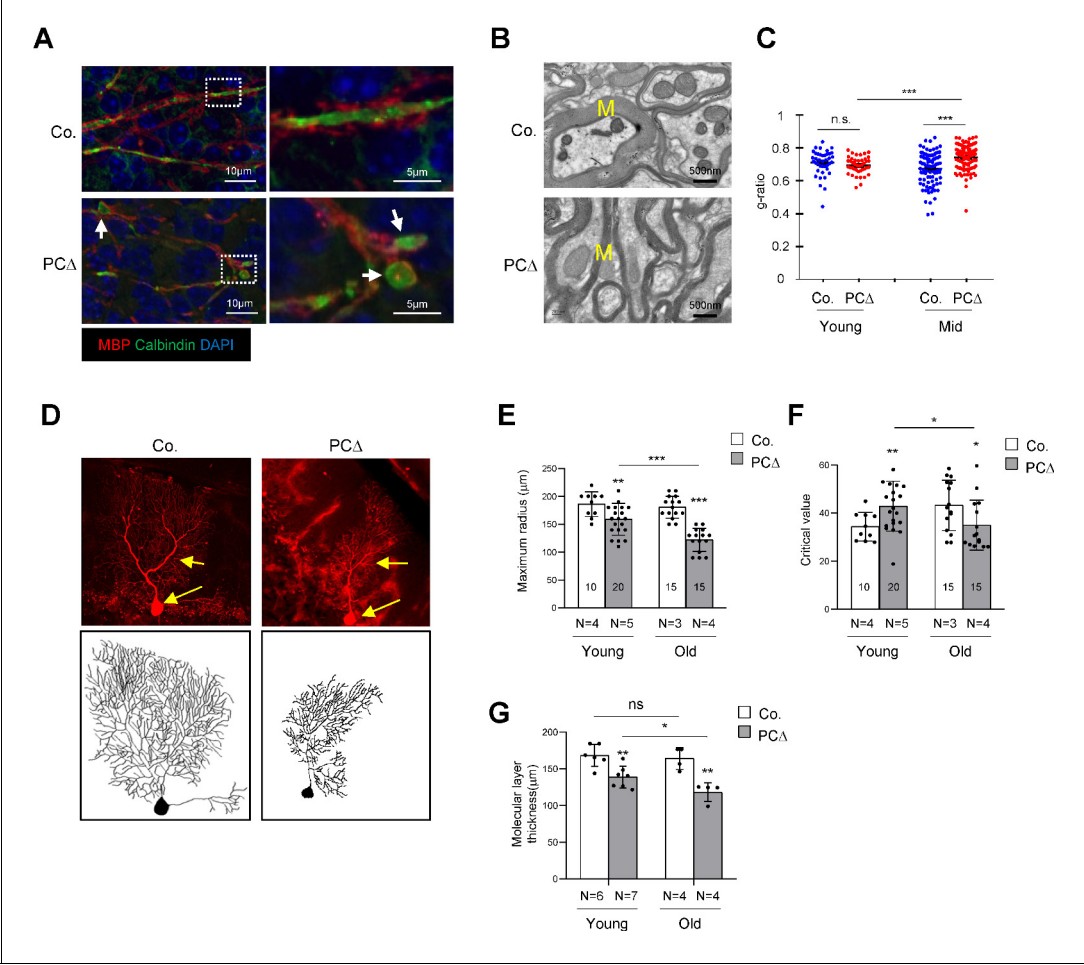

**Figure 2.** Deletion of Trrap in Purkinje cells leads to defects in their axons and dendrites. (A) Cerebellar sections from 1-month-old mice were stained with antibodies against Calbindin (green, Purkinje cells) and myelin-binding protein (MBP, red, Myelin sheets) and counterstained with DAPI. White arrows indicate axonal swellings. (B) Representative images of electromicrographs showing axon myelination in the cerebellar white matter of 6-month-old control and Trrap-PCΔ mice. M: Myelin sheet. (C) The quantification of the myelination index at the indicated ages by g-ratio, which is measured by ImageJ as ag-factor (the square-root of the area of the inner surface of an axon divided by the area of the outer surface including the myelin). Thus, a high g-ratio indicates a low myelination index. (D) Single Purkinje cells were analyzed by tracing the expression of the Confetti transgene (RFP). Representative Purkinje cell images of maximum intensity projection (MIP) from Z-stacks (upper panel) of 10-month-old mice are shown after reconstruction (lower panel) based on RFP expression in Trrap-PCΔ mice. (E) The quantification of the maximum radius after Sholl analysis, at the indicated ages, demonstrating that Purkinje cells retract their dendrites in Trrap-PCΔ mice. Young: 1–4 months; old: 10 months. (F) The graph shows the critical value measured by the Sholl analysis, at the indicated ages, indicative of the complexity of Purkinje cells. (G) The quantification of the molecular layer thickness of all cerebellar lobes. Young: 1–4 months; old: 10 months. Co.: control; PCΔ:Trrap-PCΔ. N: the number of mice analyzed. The numbers inside the bars indicate the number of cells analyzed. Mean ± standard error of the mean is shown. Student's t-test or one-way ANOVA was performed for statistical analysis. n.s.: not significant. *$p \leq 0.05$, **$p \leq 0.01$, ***$p \leq 0.001$.

The online version of this article includes the following figure supplement(s) for figure 2:

**Figure supplement 1.** Sholl analysis of degeneration of Purkinje cells of Trrap-PCΔ mice.

both brain regions. Gene ontology (GO) analyses of the common DEGs in RNA-seq data sets of both the cortex and striatum revealed alterations in multiple signaling pathways important for neuronal processes (*Figure 3D*, *Source data 1A*, *Supplementary file 1*). Intriguingly, about a half of the Top50 pathways were linked with microtubule dynamics and its related cellular processes (*Figure 3D*).

Whole proteomic analysis of P10 Trrap-FBΔ cortices identified 122 out of 6919 proteins to be significantly altered after Trrap deletion (cutoff, $q < 0.1$) (*Figure 3B*, *Figure 3—figure supplement 1b*, *Supplementary file 2*). Notably, a comparison between RNA-seq and proteomics data sets showed

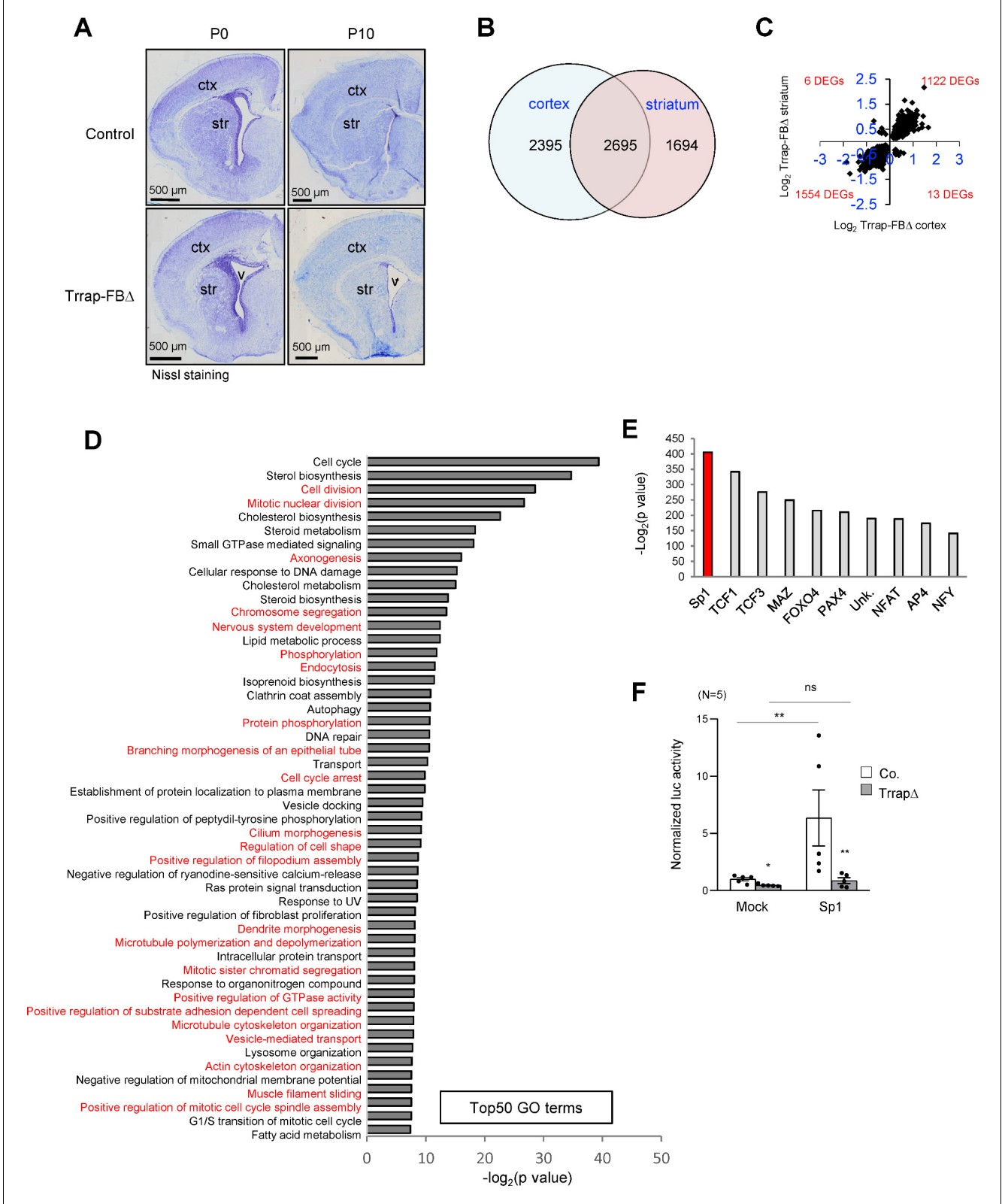

**Figure 3.** Deletion of Trrap in pyramid neurons of the forebrain results in a progress degeneration of the cortex and striatum. (**A**) Nissl staining of the coronal session of Trrap-FBΔ brain at postnatal day 0 (P0) and 10 (P10). Ctx: cortex; str: striatum; v: ventricle. (**B**) The Venn diagram depicts the overlap between the differentially expressed genes (DEGs) measured by RNA-seq in the cortex and striatum. The numbers refer to the DEGs in the indicated data sets. (**C**) Log$_2$ of the fold changes of the 2695 common DEGs in Trrap-FBΔ cortex and striatum. (**D**) Top50 GO terms of the 2695 overlapping hits

*Figure 3 continued on next page*

*Figure 3 continued*

identified in the RNA-seq data set of the cortex and striatum. Note that microtubule dynamics related processes are highlighted in red. (E) Transcription factor binding site (TFBS) enrichment analysis of the 1261 common DEGs in aNSCs, the cortex, and the striatum identified by RNA-seq. (F) Luciferase assays using a Sp1-responsive construct. The graph shows the luciferase activity normalized by Bradford assay. N: the number of cell lines analyzed; Mock: empty vector, Sp1: overexpression; luc: luciferase. Co.: control; aNSCsΔ: Trrap-aNSCsΔ. n: the number of cell lines analyzed. Mean ± standard error of the mean is shown. Unpaired t-test was performed for statistical analysis. n.s.: not significant; *p≤0.05, **p≤0.01.

The online version of this article includes the following figure supplement(s) for figure 3:

**Figure supplement 1.** Trrap deletion leads to transcriptome and proteome changes.

**Figure supplement 2.** Comparative Omics analysis of Trrap deleted aNSCs with Trrap-FBΔ brains.

---

that 85% of the proteins altered by Trrap deletion, i.e., 33 upregulated and 71 downregulated genes, were altered in the same way at the RNA level, which resembled 100% directionality (*Figure 3B*, *Figure 3—figure supplement 1c*, *Supplementary file 2*). These results indicate that most changes in the proteome were due to changes in the transcriptome.

## Trrap deletion alters Sp1 pathway

TRRAP is a cofactor interacting with various transcription factors and recruits HAT activity to their target gene promoters. To understand through which transcription factors Trrap was mediating its function, we performed a transcription factor binding site (TFBS) enrichment analysis on the common DEGs in Trrap-deleted cortices and striata. We found that most of these transcriptional changes after Trrap deletion were mediated mainly by limited transcription factors, among which transcription factors Sp1 and TCFs appeared to be the most relevant upstream factors (*Figure 3E*, *Supplementary file 3*). They mediated Trrap-dependent changes not only upstream of the common DEGs, but also upstream of the DEGs from each data set of cortices and striata (*Supplementary file 3*). Transcription factors TCF1, TCF3, and NFAT are known effectors of the Wnt signaling pathways (*Cadigan and Waterman, 2012*) and were indeed downstream of the Sp1-mediated transcription regulation upon Trrap deletion (*Supplementary file 1*). Hence, our data suggest that Sp1 is likely the main regulator for all these changes in Trrap-deleted brains.

## The Sp1 pathway is a conserved transcriptomic network in Trrap-deleted neural cells

Sp1 is a key transcription factor capable of regulating many cellular processes, including proliferation, cell differentiation, apoptosis, immune responses, DNA damage responses, and chromatin remodeling (*Li and Davie, 2010*). We attempted to analyze the transcriptional activity of Sp1 in the absence of Trrap. To achieve this, we had to adopt an in vitro culture approach and used replicating adult neural stem cells (aNSCs) from *Trrap*^f/f^ mice expressing the CreER^T2^ transgene (*Rosa26-CreER^T2^ Gt(ROSA)26Sor^tm1(cre/ERT2)Tyj^*) (designated as Trrap-iΔ). Addition of 4-hydroxytamoxifen (4-OHT) in cultured Trrap-iΔ NSCs induced an efficient deletion of Trrap (*Figure 3—figure supplement 2a,b*). We first validated the transcriptome of aNSCs in comparison with that of Trrap-FBΔ cortex and striatum and found 1261 common DEGs among these three samples. The directionality is very conserved in 93.1% between the cortex and aNSCs (*Figure 3—figure supplement 2c,d*, *Supplementary file 1*). Intriguingly, TFBS analysis revealed a remarkable commonality of DEGs among Trrap-deleted cortices, striata, and aNSCs as the transcriptional targets of Sp1 (*Figure 3—figure supplement 2e*, *Supplementary file 3*). Thus, Trrap-HAT regulates a very conserved transcriptomic network in different brain regions as well as in NSCs. Using these cells, we then performed a luciferase reporter assay to investigate whether Sp1 is directly regulated by Trrap and detected a dramatic decrease in Sp1 activity in Trrap-deleted NSCs, compared to control cells. Strikingly, ectopic overexpression of Sp1 in Trrap-deleted cells failed to activate the Sp1-reporter (*Figure 3F*). These data indicate that Trrap is indeed required for Sp1-mediated transcriptional activation.

## Alteration of Sp1-targets after Trrap deletion

RNA-seq analysis suggests that Trrap ablation leads to changes of the Sp1-dependent transcription regulation of its downstream targets in various neurological processes (*Source data 1A*). We then

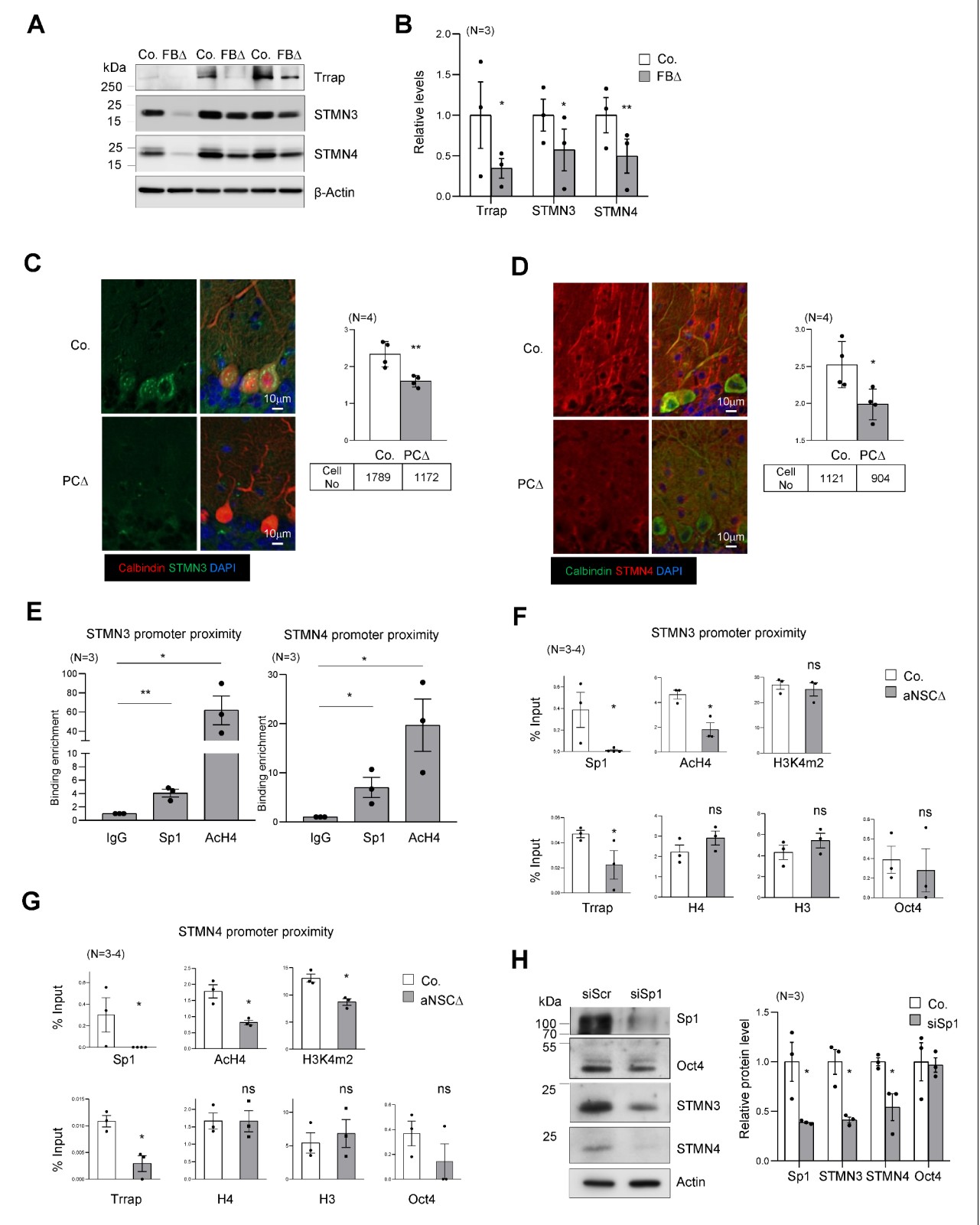

**Figure 4.** Trrap regulates the expression of STMNs via Sp1. (**A**) Western blot analysis of the Trrap deletion and expression of STMNs in the forebrain of indicated genotype at postnatal day 10 (P10). β-actin is a loading control. Co: control; FBΔ: Trrap-FBΔ. (**B**) The quantification of the expression of the indicated proteins in mutant forebrains measured by western blots are related value to adjacent controls after normalization to β-actin. N: the number of mice analyzed. The error bar presents the standard error. Paired t-test was used for statistical analysis. *p≤0.05, **p≤0.01. (**C and D**) Sagittal sections

*Figure 4 continued on next page*

*Figure 4 continued*

of 4-month-old Trrap-PCΔ mice were stained against STMN3 (green, **C**) and STMN4 (green, **D**), the Purkinje cell marker Calbindin (red) and counterstained with DAPI (blue). The right panel shows the average intensity of STMN3 or STMN4 in Purkinje cells normalized by the intensity in the neighboring cells (not Trrap-deleted). n: the number of cells analyzed; N = 4 mice analyzed. (**E**) ChIP analysis on the STMN3 and STMN4 promoters in control striata using antibodies against Sp1, AcH4, and IgG. qPCR analysis was performed to quantify the binding of the indicated factors to the promoter. The binding enrichment was calculated as fold enrichment over IgG. N = 3 mice analyzed. The primers that contain the Sp1 site for ChIP assays are marked in *Figure 4—figure supplement 1g*. (**F and G**) ChIP analysis on the proximity of STMN3 (**F**) and STMN4 (**G**) promoters in control and TrrapΔ aNSCs. Protein binding value is presented in percentage of input. The large error bars in Oct4 ChIP are due to an inclusion of a high value from one pair of samples. N = 3–4 mice analyzed. (**H**) Western blot analysis of STMN3 and STMN4 expression after siRNA-mediated knockdown of Sp1 in aNSCs. Oct4 is an Sp1 independent transcription factor control and β-actin controls loading. (**C–H**) Mean ± standard error of the mean is shown. Unpaired t-test was performed for statistical analysis. *p≤0.05, **p≤0.01.

The online version of this article includes the following figure supplement(s) for figure 4:

**Figure supplement 1.** Expression and HAT binding analysis of STMNs in Trrap-FBΔ brains.

---

compared the Sp1 targets identified in our transcriptome (DEGs) analyses (*Supplementary file 1*) with Sp1 targets identified by ChIP-seq from the Harmonizome database (*Rouillard et al., 2016*) (https://amp.pharm.mssm.edu/Harmonizome/gene_set/SP1/ENCODE+Transcription+Factor+Targets). We found that a high degree of the DEGs from Trrap-deleted brains as well as aNSCs were putative Sp1 targets (*Figure 3—figure supplement 2f*, *Supplementary file 4*). GO analysis of the common DEGs revealed that more than 30% of the Top50 pathways under the control of Sp1 were linked with microtubule dynamics-related cellular processes (*Figure 3—figure supplement 2g*, *Supplementary file 4*).

To further examine how Trrap affects HATs at target gene promoters, we performed a ChIP-seq study of acetylated histones H3 and H4 using Trrap-deleted aNSCs. Depletion of Trrap led to a downregulation of 2274 AcH3 and 1355 AcH4 peaks that were associated with coding genes (equivalent to 10% of the most depleted regions in TrrapΔ versus controls) (*Figure 4—figure supplement 1a*, *Supplementary file 5*). Only 12.6% and 10.2% respectively of these depleted peaks correlated with changes in the RNA level of the corresponding genes in Trrap-FBΔ brains (*Figure 4—figure supplement 1a*, *Supplementary file 5*). ChIP-seq analyses revealed no significant difference of AcH3 on Sp1-site between control and Trrap-deleted aNSCs, whereas H4Ac on Sp1-site in control (mean = 12.37) is slightly lower than in Trrap mutants (mean = 12.54) (*Figure 4—figure supplement 1b*). Interestingly, the downregulated genes had a significantly lower acetylation level of H3 and H4 in the Sp1 promoter area in Trrap deleted cells (*Figure 4—figure supplement 1c*). Twenty-two genes exhibited downregulated histone H3 and H4 acetylation and were also downregulated in the RNA-seq from brains. Among them, 11 genes were Sp1 targets according to the Harmonizome database (*Supplementary file 5*). The microtubule dynamics proteins STMN3 (SCLIP) and STMN4 (RB3) are of special interest (*Figure 4—figure supplement 1d,e*), because microtubule dynamics have been proposed to be involved in brain homeostasis (*Chauvin and Sobel, 2015*; *Dubey et al., 2015*) and defects in microtubule dynamics often cause axonal swellings and dendrite retraction in postmitotic neurons (*Dubey et al., 2015*; *Voelzmann et al., 2016*). Together with the high incidence of the dysregulated processes associated with microtubule dynamics, which are regulated by Sp1 (*Figure 3E*, *Figure 3—figure supplement 2f*), the finding of these two microtubule destabilizing proteins postulates this particular cellular process as the main route affected by Trrap deletion in the brain.

## Trrap-HAT mediates Sp1 transcriptional control of microtubule dynamic genes

The microtubule destabilizing proteins STMN3 and STMN4 are members of the Stathmin protein family (*Chauvin and Sobel, 2015*). STMNs 3 and 4 were found within the Top30 changes in our RNA-seq and proteomics data sets of brain samples (*Source data 1B and C*, *Supplementary file 2*). qPCR analysis confirmed a great downregulation of these genes in Trrap deleted cells of forebrain tissues (*Figure 4—figure supplement 1f*). Western blot analysis also confirmed a great reduction of both STMN3 and STMN4 proteins in Trrap-deleted forebrains at P10 (*Figure 4A,B*). We next turned our analysis to our neurodegeneration model Trrap-PCΔ mice. Co-staining of STMN3 or STMN4 with calbindin in the brain sections detected a significant decrease of both proteins in Trrap-deleted

Purkinje cells compared to controls (*Figure 4C,D*). To explore the mechanism, ChIP-seq was performed and showed that the level of AcH3 and AcH4 at the promoters of these genes was greatly reduced (*Figure 4—figure supplement 1g*). To validate the RNA-seq and ChIP-seq data, we performed ChIP experiments in brain samples and found both Sp1 binding and histone H4 acetylation were enriched at the STMN3 and STMN4 promoters in controls (*Figure 4E*). Upon Trrap deletion there was a dramatic decrease in Sp1 binding, as well as in the acetylation of histone H4 in the STMN3 and STMN4 promoters (*Figure 4F,G*). We also noted that Trrap deletion did not change the H3K4$^{m2}$ level in the STMN3 promoter, yet decreased mildly in the STMN4 promoter. Moreover, Trrap deficiency did not compromise binding of Sp1-unrelated transcription factor Oct4 at the promoter proximities of these STMNs genes (*Figure 4F,G*). These data together indicate an essentiality of Trrap for loading Sp1 and HATs to the promoters of these Sp1 target genes. Furthermore, siRNA-mediated Sp1 knockdown indeed decreased expression of STMN3 and STMN4 proteins (*Figure 4H*). These results demonstrate that the Trrap-HAT-Sp1 axis directly controls the expression of STMN3 and STMN4 in neurons.

## Functional test of STMNs in neuronal defects by Trrap deficiency

Stathmin family proteins STMN3 and STMN4 are mostly or exclusively expressed in the nervous system where they control microtubule dynamics, an essential process for neuronal differentiation, morphogenesis, and functionality (*Chauvin and Sobel, 2015*; *Dubey et al., 2015*). To investigate if Trrap deficiency especially affecting neuronal homeostasis was indeed mediated by STMNs, first we knocked down Trrap by siRNAs in primary neurons isolated from wild-type E16.5 cortex (*Figure 5—figure supplement 1a*), co-transfected with GFP or STMNs 3- or 4-expressing vectors at day 6 in vitro culture (DIV6), designated day 0 post transfection (DPT0), and analyzed the neuronal phenotype at DPT6 (*Figure 5—figure supplement 1b*). Immunofluorescence analysis revealed that the Trrap knockdown evidently reduced the neurite length and the branching numbers of neurons (*Figure 5A–C*). This was also confirmed by the IncuCyte assay at DPT6 (*Figure 5—figure supplement 1c,d*), which allows scoring a large number of cells. These findings indicate that Trrap deficiency also compromises neuronal arborization in vitro. To examine whether the downregulation of STMNs is indeed responsible for the neuropathies of Trrap deleted neurons, we ectopically expressed STMN3 in Trrap knockdown neurons (*Figure 5—figure supplement 1e*). Intriguingly, ectopic expression of STMN3 is sufficient to rescue these neuronal defects caused by Trrap knockdown (*Figure 5A,B*). Similarly, ectopic expression of STMN4 also corrected the neurite length and branching defects in Trrap knockdown primary neurons (*Figure 5A,C*). Interestingly, we note a co-upregulation of both STMNs when either STMN3 or STMN4 was overexpressed (*Figure 5—figure supplement 1e*), which suggests a co-stabilization or cooperative action of both STMNs in microtubule dynamics in the brain; yet the underlying mechanism requires further investigation. Taken together, these experiments demonstrate that Trrap prevents neuropathy by regulating Stathmin associated with microtubule dynamics.

## Discussion

The maintenance of neuron function and numbers are important for proper adult brain homeostasis. A loss of control of these processes prompts age-related neurological deficits, including neurodegeneration. Various mechanisms, including protein folding/stability, neuroinflammations, or DNA damage accumulation, have been implicated in the maintenance of brain homoeostasis and functionality. The acetylation modulations of proteins have been proposed to be important for the maintenance and function of neural cells (*Tapias and Wang, 2017*) as well as in neurodegenerative disorders, including HD, AD, PD, and ALS, yet through different mechanisms (*Cobos et al., 2019*). These studies highlight the involvement of HATs in the etiology of neurodegenerative processes. Despite the assumption that HAT/HDAC conducts a general regulatory function in transcription, how acetylation and deacetylation modulate the functionality, fitness, and even the survival of adult neurons is largely unknown. The current study shows that the HAT cofactor TRRAP is vital for preventing neurodegeneration of the Trrap-PCΔ mouse model. Trrap is an essential gene in proliferating cells because a Trrap null mutation causes lethality in cells and mice (*Herceg et al., 2001*). Unexpectedly, the deletion of Trrap in postmitotic neural cells (i.e., Purkinje neurons) is compatible with life, but elicits a full range of age-dependent neurodegenerative symptoms – axonal

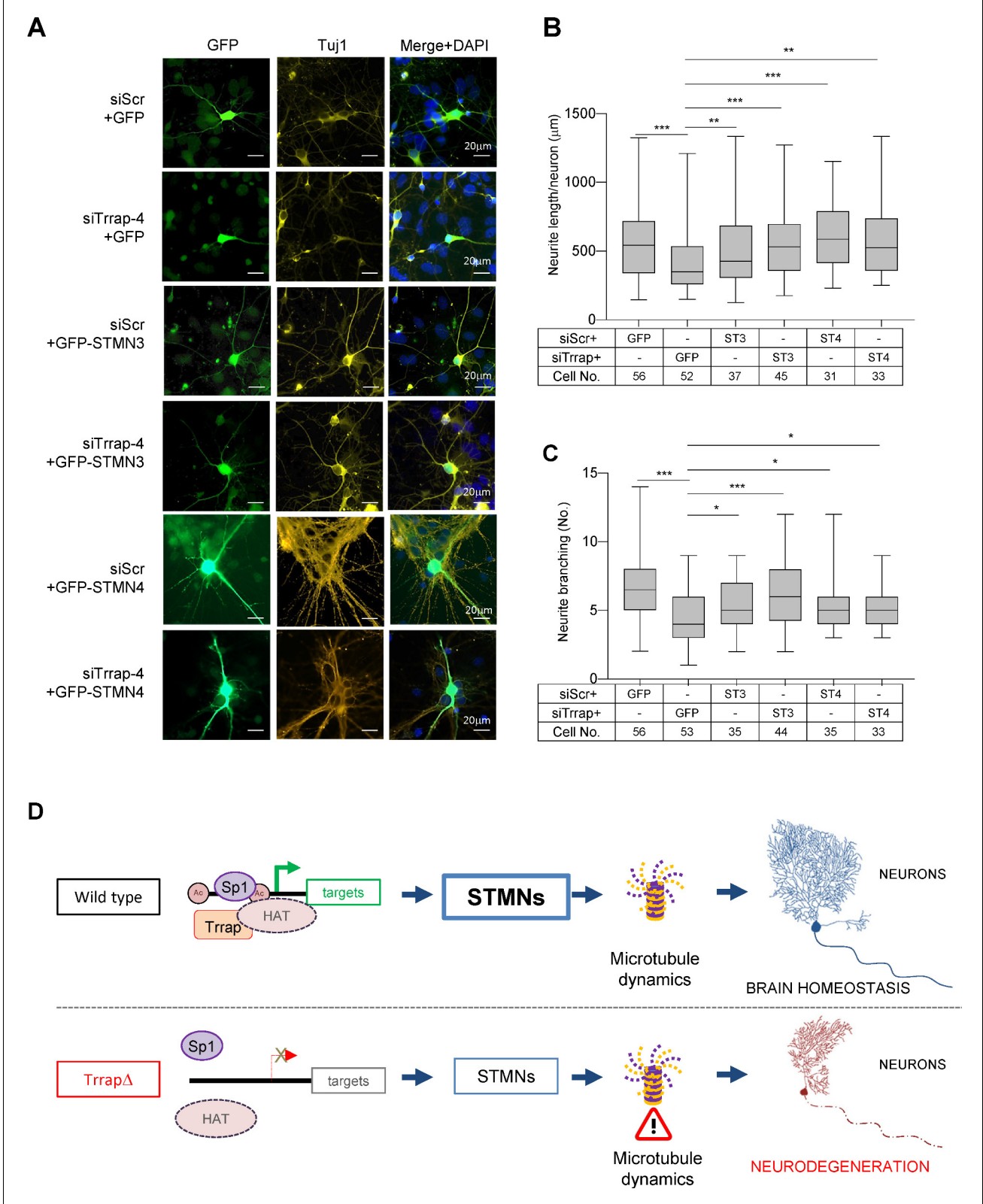

**Figure 5.** Trrap deletion causes neuronal defects in vitro that can be rescued by ectopic expression of STMN3. (**A**) Immunofluorescent images of primary neurons isolated from E16.5 forebrains at 6 days post co-transfection of siRNA (siScramble, siTrrap-4) with GFP, or with the GFP-STMN3, or with GFP-STMN4 expressing vector. (**B**) The neurite length after the Trrap knockdown and rescue by the STMN3 or STMN4 overexpression was analyzed at 6 days post co-transfection of the indicated siRNA with the GFP-, STMN3-, or STMN4-expressing vector. The neurite length is measured with NeuronJ

*Figure 5 continued on next page*

*Figure 5 continued*

(ImageJ plug-in). Only GFP-positive neurons (indicative of transfection) were analyzed. Each bar represents the data from four to six mouse embryos; the experiments were repeated more than three times. Unpaired t-test was performed for statistical analysis. **$p \leq 0.01$, ***$p \leq 0.001$, n.s., not significant. (C) The neurite branching after the Trrap knockdown and rescue by STMN3 or STMN4 overexpression was analyzed at 6 days post co-transfection of the indicated siRNA with the GFP, STMN3-, or STMN4-expressing vector. The neurite length is measured with NeuronJ (ImageJ software). Only GFP-positive neurons were scored and are shown. Each bar represents the data from four to six mouse embryos; the experiments were repeated more than three times. Unpaired t-test was performed for statistical analysis. *$p \leq 0.05$, **$p \leq 0.01$, ***$p \leq 0.001$, n.s., not significant. (D) Working model of Trrap-HAT-Sp1 in brain homeostasis and neurodegeneration. The Trrap deletion compromises HAT to acetylate histones resulting in insufficient binding of Sp1 and the subsequent downregulation of target genes involved in microtubule dynamics (STMNs). The dysregulation of STMNs provokes the axonal swelling, declines of neurite lengths and branching of postmitotic neurons, ultimately, leading to defective neuronal homeostasis and neurodegeneration.

The online version of this article includes the following figure supplement(s) for figure 5:

**Figure supplement 1.** Ectopic expression of STMNs rescues defects of Trrap knockdown neurons.

---

demyelination, dendrite retraction, progressive neuronal death, reactive astrogliosis, and the activation of microglial cells. Trrap deletion in non-dividing neurons, avoiding lethality, allows the identification of Sp1 as a specific master regulator, which is under the control of Trrap-HAT, to ensure proper neuronal arborization and to prevent neuron loss (*Figure 5D*). Although our omics studies detect a range of genes that have been altered, microtubule dynamics seems to be particularly vulnerable to Trrap deletion, because the major changes in the Trrap deleted brains are the processes involving microtubule dynamics and that ectopic expression of microtubule destabilizing proteins STMNs can largely ameliorate the arborization defects of Trrap-deficient neurons.

The TFBS enrichment analysis of these commonly dysregulated genes in Trrap-deleted brains points to selective transcription programs of the Trrap-HAT downstream. Our integrated omics analyses revealed a remarkable commonality of Trrap-HAT-regulated genes via Sp1 in the neurons of the cortex and striatum, and even with neural stem cells. Intriguingly, the Sp1 pathway is on the top of the changes by Trrap deletion and Trrap is required for the Sp1 transcriptional activity. The action of Trrap-HAT in postmitotic neurons is to regulate, via Sp1, the expression of neuroprotection and brain homeostasis genes, among which microtubule dynamics is most affected (*Dubey et al., 2015*; *Noelanders and Vleminckx, 2017*). Although we have not completely confirmed that all these molecular pathways governed by Trrap-HAT-Sp1 in pyramidal neurons (Trrap-FBΔ mice) would be identical in Purkinje neurons (due to the technical limitation), STMN3/4 were indeed downregulated in Trrap-PCΔ models. In agreement with the ChIP data, knockdown of Sp1 repressed STMN3/4 expression. Thus, it is likely that the Sp1-mediated specific transcriptome could also function in the prevention of neurodegeneration.

Sp1 is a master transcription factor binding to the GC box of promoters and can also regulate by itself. It has many downstream target genes, among which the regulation of cancer and cell proliferation have been well studied (*Li and Davie, 2010*; *Vizcaíno et al., 2015*; *Suske, 2017*). However, the upstream regulatory mechanism of Sp1 has not been defined previously. Here we show that Trrap-HAT is upstream of Sp1 and has a specific regulatory role in the Sp1-mediated transcription. Although Sp1 has been reported to bind to the promoter of some genes in neural cells, such as Slit2 (*Saunders et al., 2016*), P2 × 7 (*García-Huerta et al., 2012*), and Reelin (*Chen et al., 2007*), it has not been linked directly with neural development and degeneration. Our transcriptome analyses reveal that many neurological processes are indeed regulated by Sp1 downstream targets, many of which are connected with microtubule dynamics (see *Figure 3D*, *Figure 3—figure supplement 2g*, *Source data 1A*). For example, Trrap facilitates Sp1 binding to the gene loci of the microtubule destabilizing proteins STMN3 and STMN4. These findings are particularly interesting, because Sp1 has been implicated in neurodegeneration disorders; yet previously published data are often controversial. A GWAS analysis detected Sp1 among candidates mediating transcriptional activity changes in AD and PD patients (*Ramanan and Saykin, 2013*). Sp1 seems to prevent neurotoxicity in HD (*Dunah et al., 2002*), whereas others showed that a downregulation is protective in HD development (*Qiu et al., 2006*). Sp1 is found upregulated in AD patients and also in an AD mouse model (*Citron et al., 2008*); however, when it was chemically inhibited, memory deficits were even enhanced in AD transgenic mice (*Citron et al., 2015*), ruling out an instrumental role of Sp1 in AD. These controversial findings are perhaps not surprising, because the expression changes of the Sp1

gene and protein can be regulated by multiple mechanisms, such as transcriptional regulation, epigenetics, and posttranscriptional modifications (*Li and Davie, 2010*), or can be secondary to the manifestation of disease processes in a very heterogenous genetic background in human studies.

Our analyses identify the Trrap-HAT-Sp1 axis as a specific regulator of the microtubule dynamics process. Microtubules dynamics, coordinated, but mainly, by a destabilizing processing of microtubules, is a crucial regulator of neurite outgrowth, the maintenance of neuronal morphology and cargo transport along axons, and thereby of brain homeostasis (reviewed in *Chauvin and Sobel, 2015*; *Voelzmann et al., 2016*). The destabilizing factors STMNs play an important yet distinct function in neuronal homeostasis. Defects in microtubule dynamics can cause axonal swellings and dendrite retraction; both processes of neurodegeneration and the misregulation of STMNs have been associated with neuron abnormalities (*Dubey et al., 2015*; *Voelzmann et al., 2016*). Trrap deletion greatly downregulates STMN3 and STMN4 in Purkinje cells and also in the forebrains. This finding is particularly interesting because that STMN3 and STMN4 express preferentially high during adult life than STMN1 and STMN2 (SCG10), suggesting their important functions in neuronal homeostasis (*Chauvin and Sobel, 2015*; *Voelzmann et al., 2016*; *Ozon et al., 1998*). The repression of STMN3 suppresses the Purkinje cell neurite growth and arborization; when overexpressed, it promotes dendritic elongation and branching (*Poulain et al., 2008*). In contrast to STMN3, the role of STMN4 in neuronal morphogenesis and arborization has not been extensively studied. Moreover, STMN1 knockout mice showed an age-dependent axonopathy, characterized by a progressive degeneration of axons and demyelination (*Liedtke et al., 2002*). Also, an overexpression of STMN2 has been shown to enhance neurite outgrowth by favoring microtubules disassembly (*Morii et al., 2006*). Therefore, it is plausible that downregulation of STMN3 and STMN4 is responsible for axonal swellings and the dendrite retraction of *Trrap*-deleted Purkinje cells. This conclusion is further supported by our genetic knockdown and rescue experiments showing that an ectopic expression of either STMN3 or STMN4 can effectively rescue the defects of neurite length and branching of primary cortical neurons, which are caused by Trrap knockdown. However, an in vivo rescue of neuron loss in Trrap-PCΔ mice has been impossible due to a technique limitation. Nonetheless, our data demonstrate an instrumental role of the Trrap-Sp1-mediated transcriptional control of microtubule dynamics, likely via STMNs, in neuronal morphogenesis and preventing neurite retraction.

Very recently, human patients carrying de novo mutations of TRRAP are reported to be associated with neuropathological symptoms, such as ID, ASD, and epilepsy (*Cogné et al., 2019*; *Mavros et al., 2018*). Although molecular pathways have not been investigated in these genetic studies, our findings hint at the Sp1 transcription regulation network by HAT being a plausible mechanism. This current study is the first report demonstrating that the general transcriptional regulator Sp1 is specifically under the control of Trrap-HAT. Our findings propose that the Trrap-HAT-Sp1 axis orchestrates the program of neuronal microtubule dynamics and neuron morphogenesis and, possibly, neuron survival.

## Materials and methods

**Key resources table**

| Reagent type (species) or resource | Designation | Source or reference | Identifiers | Additional information |
|---|---|---|---|---|
| Gene (*M. musculus*) | Trrap | Genebank | MGI:MGI:2153272 | |
| Gene (*M. musculus*) | Sp1 | Genebank | MGI:MGI:98372 | |
| Gene (*M. musculus*) | STMN3 | Genebank | MGI:MGI:1277137 | |
| Gene (*M. musculus*) | STMN4 | Genebank | MGI:MGI:1931224 | |

*Continued on next page*

*Continued*

| Reagent type (species) or resource | Designation | Source or reference | Identifiers | Additional information |
|---|---|---|---|---|
| Strain, strain background (*M. musculus*) | Trrapf/f; Pcp2-Cre | This paper | | Trrap deletion in Purkinje cells; *M. musculus*, male and female; Please refer to 'Materials and methods' in the paper, Section 'Mice' |
| Strain, strain background (*M. musculus*) | Trrapf/f; Camk2-Cre | This paper | | Trrap deletion in forebrain; *M. musculus*, male and female; Please refer to 'Materials and methods' in this paper, Section 'Mice' |
| Strain, strain background (*M. musculus*) | Trrapf/+; Rosa26-CreERT2 Trrapf/f; Rosa26-CreERT2 | This paper | | Trrapf/+ acts as a control to Trrapf/f Trrap deletion in adult neural stem cells; *M. musculus*, male and female; Please refer to 'Materials and methods' in this paper, Section 'Mice' |
| Strain, strain background (*M. musculus*) | B6.Cg-Tg(Thy1-Brainbow1.0)HLich/J (R26R-Confetti); Trrapf/f; Pcp2-Cre | This paper | | Tracing of the single Purkinje cells; *M. musculus*, male and female; Please refer to 'Materials and methods' in this paper, Section 'Mice' |
| Genetic reagent (*M. musculus*) | Lipofectamine 2000 | Invitrogen | Cat#: 11668027 | siTrrap and Plasmid co-transfection; *M. musculus* |
| Genetic reagent (*M. musculus*) | Lipofectamine RNAiMAX | Invitrogen | Cat#: 13778075 | siSp1 transfection; *M. musculus* |
| Cell line (*M. musculus*) | Trrap-aNSC | This paper | | Primary cell line; *M. musculus*; Please refer to 'Materials and methods' in this paper, Section 'aNSC cell culture'. |
| Cell line (*M. musculus*) | E16.5 cortical neuron | This paper | | Primary cell line; *M. musculus*; Please refer to 'Materials and methods' in the paper, Section 'Isolation and culture of murine primary neurons'. |
| Cell line (*M. musculus*) | Neuro-2a Neuroblastoma cells | PMID:4534402 | ATCC CCL-131 | Cell line; *M. musculus* |
| Transfected construct (*M. musculus*) | ON-TARGETplus siRNA Reagents -Mouse (siScramble) | Horizon Discovery | Cat#: D-001810-10-05 | UGGUUUAC AUGUCGACUAA; *M. musculus* |
| Transfected construct (*M. musculus*) | siTrrap-1 | Horizon Discovery | Cat#: LQ-051873-01-0005 | CAAAAGUAG UGAACCGCUA; *M. musculus* |
| Transfected construct (*M. musculus*) | siTrrap-2 | Horizon Discovery | Cat#: LQ-051873-01-0005 | CCUACAUUG UGGAGCGGUU; *M. musculus* |

*Continued on next page*

*Continued*

| Reagent type (species) or resource | Designation | Source or reference | Identifiers | Additional information |
|---|---|---|---|---|
| Transfected construct (*M. musculus*) | siTrrap-3 | Horizon Discovery | Cat#: LQ-051873-01-0005 | GCCAACUGUC AGACCGUAA; *M. musculus* |
| Transfected construct (*M. musculus*) | siTrrap-4 | Horizon Discovery | Cat#: LQ-051873-01-0005 | CGUACCUGG UCAUGAACGA; *M. musculus* |
| Antibody | Anti-Calbindin (Mouse Monoclonal) | Sigma | Cat#:C9848 RRID:AB_476894 | IF:1:300 WB: 1:1000 |
| Antibody | Anti-GFAP (Mouse Monoclonal) | Agilent | Cat#:G3893 RRID:AB_477010 | IF:1:300 WB: 1:1000 |
| Antibody | Anti-MBP (Mouse Monoclonal) | Millipore | Cat#:MAB384 RRID:AB_240837 | IF:1:300 |
| Antibody | Anti-GFP (Rabbit Monoclonal) | Cell Signaling Technology | Cat#:2956 RRID:AB_1196615 | IF: 1:200 |
| Antibody | Anti-GFP (Mouse Monoclonal) | Santa Cruz | Cat#:sc-390394 | IF:1:200 WB: 1:400 |
| Antibody | Anti-Sp1 (Mouse Monoclonal) | Santa Cruz | Cat#:sc-17824 RRID:AB_628272 | IF: 1:50 |
| Antibody | Anti-Sp1 (Rabbit Polyclonal) | Millipore | Cat#:07–645 RRID:AB_310773 | WB:1:1000 ChIP: 1:80 |
| Antibody | Anti-STMN3 (Rabbit Polyclonal) | Proteintech, | Cat#:11311–1-AP RRID:AB_2197399 | IF:1:100 WB:1:1000 |
| Antibody | Anti-STMN4 (Mouse Monoclonal) | Santa Cruz | Cat#:sc-376829 | IF:1:100 WB:1:1000 |
| Antibody | Anti-Tuj1 (Mouse Monoclonal) | Covance | Cat#: MMS-435P RRID:AB_2313773 | IF:1:400 |
| Antibody | Anti-CNPase (Mouse Monoclonal) | Sigma | Cat#: SAB4200693 | IF:1:1000 |
| Antibody | Anti-Galectin3 (Rat Monoclonal) | eBioscience | Cat#:14-5301-82 RRID:AB_837132 | WB:1:1000 |
| Antibody | Anti-a-tubulin (Mouse Monoclonal) | Sigma | Cat#:sc-32293 RRID:AB_628412 | WB: 1:5000 |
| Antibody | Anti-TRRAP (Mouse) clone TRR-2D5 | Euromedex | ID: IG-TRR-2D5 | WB:1:1000 |
| Antibody | Anti-TRRAP (Mouse) clone TRR-1B3 | Euromedex | ID: IG-TRR-1B3 | ChIP: 1:40 |
| Antibody | Anti-β-actin (Mouse Monoclonal) | Sigma | Cat#:A5441 RRID:AB_476744 | WB:1:3000 |
| Antibody | Anti-AcH3 (Rabbit Polyclonal) | Millipore | Cat#:06–599 RRID:AB_2115283 | ChIP: 1:150 |

*Continued on next page*

*Continued*

| Reagent type (species) or resource | Designation | Source or reference | Identifiers | Additional information |
|---|---|---|---|---|
| Antibody | Anti-AcH4 (Rabbit Polyclonal) | Millipore | Cat#:06–866 RRID:AB_310270 | ChIP: 1:150 |
| Antibody | Anti-H3K4me2 (Rabbit Polyclonal) | Abcam | Cat#: ab7766 RRID:AB_2560996 | ChIP: 1:100 |
| Antibody | H3 (Rabbit Monoclonal) | Abcam | Cat#: ab1791 RRID:AB_302613 | ChIP: 1:150 |
| Antibody | H4 (Rabbit Polyclonal) | Abcam | Cat#: ab7311 RRID:AB_305837 | ChIP: 1:150 |
| Antibody | Oct-4 (Rabbit Monoclonal) | Cell Signaling | Cat#: 2840 RRID:AB_2167691 | ChIP: 1:80 WB:1:1000 |
| Antibody | IgG (Rabbit Polyclonal) | Sigma | Cat#: I8140 RRID:AB_1163661 | ChIP: 1:1500 (2 µg antibody) |
| Recombinant DNA reagent | EF1a-GFP-P2A-STMN3-Poly(A) (plasmid) | This paper | | STMN3 overexpression plasmid; *M. musculus*; Please refer to 'Materials and methods' in this paper, Section 'Construction of STMNs expression vectors'. |
| Recombinant DNA reagent | EF1a-GFP-Poly(A)-EF1a-STMN4-Poly(A) (plasmid) | This paper | | STMN4 overexpression plasmid; *M. musculus*; Please refer to 'Materials and methods' in this paper, Section 'Construction of STMNs expression vectors'. |
| Recombinant DNA reagent | −111 hTF m3 | Addgene | Cat#: 15450 | Sp1 activity reporter; *H. sapiens* |
| Recombinant DNA reagent | pN3-Sp1FL | Addgene | Cat#: 24543 | Sp1 overexpression reporter; *H. sapiens* |
| Sequence-based reagent | Sp1 primer | PrimerBank | ID 7305515a1 | Fwd, 5'-GCCGCCT TTTCTCAGACTC-3'; Rev, 5'-TTGGGTGACT CAATTCTGCTG-3' |
| Sequence-based reagent | STMN3 primer | PrimerBank | ID 6677873a1 | Fwd, 5'-CAGCACCG TATCTGCCTACAA-3'; Rev, 5'-GTAGATGGT GTTCGGGTGAGG-3'; |
| Sequence-based reagent | STMN4 primer | PrimerBank | ID 9790189a1 | Fwd, 5'-ATGGAAGT CATCGAGCTGAACA-3'; Rev, 5'-GGGAGGCATT AAACTCAGGCA-3'. |
| Sequence-based reagent | STMN3 promoter primer | This paper | | Fwd, 5'-CTTGCTACTG CATCAGGCGA-3'; Rev, 5'-AGCCTAGGG GATCATGGGAC-3'; |
| Sequence-based reagent | STMN4 promoter primer | This paper | | Fwd, 5'-TCGCTTTGG AAACCGGACTG-3'; Rev, 5'-TTTGTTT AAAACCCCCGCCC-3'. |

*Continued on next page*

*Continued*

| Reagent type (species) or resource | Designation | Source or reference | Identifiers | Additional information |
|---|---|---|---|---|
| Commercial assay or kit | Incucyte S3 | Sartorius AG | Product Code: 4695 | For neurite detection and quantification |
| Commercial assay or kit | RNeasy Lipid Tissue Mini Kit | Qiagen | Cat #: 74804 | |
| Commercial assay or kit | RNAeasy Mini Kit | Qiagen | Cat #: 74104 | |
| Commercial assay or kit | LightCycler 480 Real-Time PCR System | Roche | Product No. 05015243001 | |
| Commercial assay or kit | RNA 6000 nano kit | Agilent | Cat #: 5067–1511 | |
| Commercial assay or kit | TruSeq Stranded mRNA Kit | Illumina | Cat #: 20020594 | |
| Commercial assay or kit | Dual-Glo Luciferase Assay System | Promega | Cat# E2920 | |
| Commercial assay or kit | QiaQuick PCR Purification Kit | Qiagen | Cat# 28106 | |
| Commercial assay or kit | Fragment Analyzer | Agilent | Cat#: M5310AA | |
| Commercial assay or kit | NextSeq500 platform | Illumina | RRID:SCR_014983 | |
| Commercial assay or kit | TruSeq ChIP Sample Preparation Kit | Illumina | Cat#: IP-202–1024 | |
| Chemical compound, drug | Epoxy resin 'Epon' | SERVA | | Glycid ether 100 for electron microscopy |
| Chemical compound, drug | cOmplete, Mini, EDTA-free | Roche | Cat#: 04693159001 | Protease Inhibitor |
| Chemical compound, drug | PhosSTOP | Roche | Cat#: PHOSS-RO | Phosphatase Inhibitor |
| Chemical compound, drug | protein-A-conjugated magnetic beads | Invitrogen | Cat#: 10003D | |
| Chemical compound, drug | protein-G-conjugated magnetic beads | Invitrogen | 10001D | |
| Chemical compound, drug | Platinum SYBR Green qPCR SuperMix-UDG | Qiagen | 11733046 | |
| Software, algorithm | NeuronJ Plug-in by ImageJ software | National Institutes of Health | | Neurite tracing and quantification |
| Software, algorithm | Fiji plugins Simple Neurite Tracing | National Institutes of Health | | Sholl analysis |
| Software, algorithm | bcl2FastQ | Illumina | RRID:SCR_015058 | Version 1.8.4 |

*Continued on next page*

*Continued*

| Reagent type (species) or resource | Designation | Source or reference | Identifiers | Additional information |
|---|---|---|---|---|
| Software, algorithm | STAR | PMID:23104886 | RRID:SCR_015899 | Version 2.5.4b; RNA sequence mapping parameters: `-alignIntronMax` 100000 `-outSJfilterReads` Unique `-outSAMmultNmax` 1 `-outFilterMismatch NoverLmax` 0.04 |
| Software, algorithm | FeatureCounts | PMID:24227677 | RRID:SCR_012919 | Version 1.5.0; parameters: metafeature mode, stranded mode '2', Ensembl 92 annotation |
| Software, algorithm | ENSEMBL annotation | PMID:31691826 | RRID:SCR_002344 | Release 92 for *Mus musculus* |
| Software, algorithm | MultiQC | PMID:27312411 | RRID:SCR_014982 | Version 1.6; RNA sequence quality assessment of the raw input data, the read mapping and assignment steps |
| Software, algorithm | R package DESeq2 | PMID:25516281 | RRID:SCR_015687 | Version 1.20.0; Analysis of differential expressed genes in pairwise comparisons. |
| Software, algorithm | R package VennDiagram | PMID:21269502 | RRID:SCR_002414 | Version 1.6.20 |
| Software, algorithm | Database for Annotation, Visualization and Integrated Discovery (DAVID) programs | https://david.ncifcrf.gov/home.jsp | | DAVID v6.7; Gene ontology (GO) and KEGG pathway enrichment analyses |
| Software, algorithm | TFBS enrichment analysis | UC San Diego, Broad Institute, USA | GSEA 4.1.0 | Based on GSEA database or Harmonizome database for Sp1 targets |
| Software, algorithm | Ingenuity Pathway Analysis (IPA) program | Qiagen | | Analysis of Sp1 targets affected by Trrap deletion |
| Software, algorithm | R package AnnotationDbi | Bioconductor | DOI: 10.18129/B9.bioc.AnnotationDbi | Version 1.42.1 |
| Software, algorithm | R package org.Mm.eg.db | Bioconductor | DOI: 10.18129/B9.bioc.org.Mm.eg.db | Version 3.6.0 |
| Software, algorithm | FastQC | Babraham Bioinformatics, UK | RRID:SCR_014583 | Version 0.11.5 |
| Software, algorithm | Bowtie | http://bowtie-bio.sourceforge.net | RRID:SCR_005476 | Version 1.1.2 |
| Software, algorithm | MACS14 | https://bio.tools/macs | RRID:SCR_013291 | |
| Software, algorithm | R | https://www.r-project.org/ | RRID:SCR_001905 | Version 3.4.4 |

*Continued on next page*

*Continued*

| Reagent type (species) or resource | Designation | Source or reference | Identifiers | Additional information |
|---|---|---|---|---|
| Other | Beam walking | Homemade | | |
| Other | Mouse Rota-rod | Ugo Basile | Cat#: 47600 | |
| Other | DAPI stain | Invitrogen | Cat#: D1306 | 1:5000 |
| Other | Bioruptor | Diagenode | N/A | Sonication |
| Other | vibrating microtome HM 650 V | Thermo Scientific Microm | | Sagittal section cutting |
| Other | Reichert Ultracut S | Leica | | Ultrathin section cutting |
| Other | JEM 1400 electron microscope | JEOL | | Electron microscopic imaging |
| Other | Orius SC 1000 CCD-camera | GATAN | | Electron microscopic imaging |
| Other | Bioanalyzer 2100 | Agilent | | Quality check and quantification of RNA |

## Mice

Mice carrying the conditional (floxed; *Trrap*^f/f^) allele (*Herceg et al., 2001*) were crossed with *Pcp2*-Cre transgenic mice (Tg(Pcp2-cre)2Mpin) (*Barski et al., 2000*), *Camk2*-Cre (Tg(Camk2a-cre/ERT2)2Gsc), or *Rosa26*-CreER^T2^ Gt(ROSA)26Sor^tm1(cre/ERT2)Tyj^, to generate mice with a specific deletion in Purkinje cells (Trrap-PCΔ) and forebrain glutamatergic neurons (Trrap-FBΔ), or an inducible deletion in all tissues (Trrap-iΔ). To trace the single cell morphology of Purkinje cells, B6.Cg-Tg(Thy1-Brainbow1.0)HLich/J (R26R-Confetti) knock-in mice were crossed with Trrap-PCΔ mice. The double-fluorescent reporter *mT/mG* knock-in mice (*Muzumdar et al., 2007*) were intercrossed with Trrap-aNSCΔ mice, to identify Trrap-deleted cells. The Trrap, Cre, mT/mG, and Confetti genotypes of mice were determined by PCR on DNA extracted from tail tissue, as previously described (*Loizou et al., 2009*). Animal experiments were conducted according to German animal welfare legislation, and the protocol is approved by Thüringen Landesamt für Verbraucherschutz (TLV) (03-042/16), Germany.

## Histology

Tissues for histology were fixed in 4% paraformaldehyde (PFA), cryoprotected in 30% sucrose and frozen in Richard-Allan Scientific Neg-50 Frozen Section Medium (Thermo Scientific, Waltham, MA, USA). The sections (thickness of 5–20 μm) were later used for immunofluorescence staining.

## Construction of STMNs expression vectors

The EF1a-GFP-P2A-STMN3-Poly(A) plasmid was generated by subcloning the RV-Cre2A-GFP (kindly provided by Xiaobing Qing) and the STMN3 protein coding region into the EF1a-GFP construct (*Li et al., 2015*). For the EF1a-GFP-Poly(A)-EF1a-STMN4-Poly(A) vector, EF1a-promoter, STMN4 protein coding region, and Poly(A) sequence were subcloned into the EF1a-GFP construct. The DNA fragments were assembled with Gibson Assembly Master Mix (New England Biolabs, Massachusetts, USA). The STMN3 and STMN4 protein coding regions were amplified from cDNA library of the murine 10 days postnatal forebrain samples.

## siRNA sequences

## Isolation and culture of murine primary neurons

Murine neurons were isolated from mouse embryos at embryonic stage E16.5 (E16.5). The cortex was removed and was first incubated with 0.05% trypsin under 37°C for 15 min. The tissue was then

mechanically disintegrated with 1 ml Eppendorf pipettes in an incubation medium (Eagle's minimal essential medium (MEM) supplemented with 1× FCS, 1× B-27 Supplements, 500 µM L-glutamine, 1 mM sodium pyruvate, 1× penicillin–streptomycin, 10 mM HEPES). The suspension was filtered through a cell strainer (40 µm porosity). After centrifugation (630 rpm for 5 min) the neurons in supernatant were seeded into poly-L-lysine coated multiple well plates at the indicated number (6 × $10^4$ cells/well in 24-well plate, 3 × $10^5$ cells/well in 6-well plate) and cultured in the Neurobasal medium (Thermo Fisher Scientific, Waltham, Massachusetts, USA) supplemented with 1 × B-27 Supplements, 500 µM L-glutamine, and 10 mM HEPES until further use.

## Motor coordination tests

Beam walking: Mice were trained to run along a 1 m long beam (3 cm thick) to their home cage. The test was performed on five consecutive days on a 2 cm thick beam, with three runs each day. The mice were video-taped and timed crossing the beam.

Rotarod test: Mice were habituated to the test situation by placing them on a rotarod (Ugo Basile, Gemonio, Italy) with constant rotation (5 rpm) for 5 min the day prior to the test. In the test phase, two trials per mouse were performed with accelerating rotation (2–50 rpm within 4 min) and maximum duration of 5 min, with the time measured until mice fell off the rod.

## N2A cell culture

N2A cells were cultured in DMEM supplemented with 1× FCS, 1× penicillin–streptomycin, and 10 mM HEPES. When the N2A culture reached ~70% confluency, the cells were trypsinized and the cell suspension was centrifuged. The cells were seeded in 1.5 × $10^5$ cells/well onto 6-well plate.

## Transfection of primary neurons or N2A cells

Primary neurons were transfected on day 6 in vitro (DIV6) and N2A cells on day 1 after passage using lipofectamin 2000 in Opti-MEM (0.4 µg plasmid + 0.8 µl lipofectamine 2000 in 100 µl Opti-MEM/well in 24-well plate, 1.2 µg plasmid + 2.4 µl lipofectamine 2000 in 300 µl Opti-MEM/well in 6-well plate, and/or 25 µM siRNA). After 30 min incubation under 37°C with the Neurobasal medium supplemented with 500 µM L-glutamine (300 µl/well in 24-well plate, 1.2 ml/well in 6-well plate), the plasmid/siRNA-Lipofectamine mix was replaced by the neuronal culture medium.

## IncuCyte quantification

The primary neuron culture was placed into Incucyte S3 (Sartorius AG, Göttingen, Germany) for imaging acquisition of phase contrast and GFP signals (10× magnification, 36 images/well in 24-well plate, and 144 images/well in 6-well plate). The image analysis and the neurite detection parameter were determined for each plate separately through IncuCyte NeuroTrack Software Module for S3 or ZOOM.

## Immunofluorescent staining and quantification

Prior to immunostaining, primary neurons on coverslips were fixed with 4% PFA and incubated with 0.7% Triton in PBS for 15 min. The fixed samples were incubated with primary antibodies under 4°C overnight. After incubation with secondary antibody in 1:5000 DAPI, the samples were conserved by ProLong Gold Antifade reagent (Thermo Fisher Scientific). Images were captured by the ApoTome microscope (Zeiss Jena, Germany) under 20× or 40× objectives. The neurite branching, neurite length, and axonal swelling were then scored with NeuronJ Plug-in by ImageJ software and validated manually.

## qRT-PCR analysis

The total RNA was isolated from tissues or aNSCs using the RNeasy Lipid Tissue Mini Kit (Qiagen) and an RNAeasy Mini Kit (Qiagen) respectively and following the manufacturer's instructions. cDNA was synthesized using the SuperScript III Reverse Transcriptase (Thermo Fischer Scientific). qPCRs were performed using Platinum SYBR Green qPCR SuperMix-UDG (Thermo Fischer Scientific) and a LightCycler 480 Real-Time PCR System (Roche). The Trrap and Actin primers used for amplification were previously described (*Tapias et al., 2014*). The remaining primer sequences were obtained from the PrimerBank (*Spandidos et al., 2010*; *Spandidos et al., 2008*; *Wang, 2003*) and were as

follows: Sp1 (PrimerBank ID 7305515a1): Fwd, 5'-GCCGCCTTTTCTCAGACTC-3'; Rev, 5'-TTGGG TGACTCAATTCTGCTG −3'; STMN3 (PrimerBank ID 6677873a1): Fwd, 5'- CAGCACCGTATCTGCC TACAA-3'; Rev, 5'-GTAGATGGTGTTCGGGTGAGG-3'; STMN4 (PrimerBank ID 9790189a1): Fwd, 5'- ATGGAAGTCATCGAGCTGAACA-3'; Rev, 5'- GGGAGGCATTAAACTCAGGCA-3'. Quantification of the qPCR data was performed by the $\Delta\Delta Cp$ method using actin as an internal control. Gene expression values were expressed relative to the gene expression in control tissues or aNSCs.

## TUNEL reaction and immunofluorescence staining in brain sections

Immunofluorescence and TUNEL staining were performed on cryosections prepared from PFA-fixed brains of the indicated ages, as previously described (Tapias et al., 2014), using the following antibodies: mouse anti-Calbindin (1:300, Sigma), rabbit anti-GFAP (1:300, Agilent, Santa Clara, USA), mouse anti-MBP (1:300, Millipore, Burlington, USA), rabbit anti-Calbindin (1:300, Swant, Marly, Switzerland), rabbit anti-GFP (1:200, Cell Signaling Technology, Danvers, USA) mouse anti-GFP (1:200, Santa Cruz), rabbit anti-Sp1 (1:50, Santa Cruz), mouse anti-STMN4 (1:100, Santa Cruz), rabbit anti-STMN3 (1:100, Proteintech, Rosemont, USA), and mouse anti-Tuj1 (1:400, Covance).

## Immunoblot analysis

Total protein lysates were prepared from brain tissue or aNSCs using the RIPA buffer (50 mM Tris-HCl, pH 7.4, 150 mM NaCl, 1% NP40, 0.25% Na-deoxycholate, 1 mM EDTA, 1 mM PMSF), and complete mini protease inhibitor cocktail (Roche Applied Science, Penzberg, Germany). N2A cells or neuron lysates were prepared as follows: the culture was treated with 0.25% trypsin for 5 min under 37°C, resuspend with DMEM medium supplemented with $1\times$ FCS and centrifuged under 1100 rpm for 5 min. The resulting pellets were washed once with PBS and lysed with the RIPA buffer. Immunoblotting was performed as described previously (Tapias et al., 2014), using the following antibodies: mouse anti-Calbindin (1:1000, Sigma), rabbit anti-GFAP (1:1000, Dako-Agilent), mouse anti-CNPase (1:1000, Sigma), rat anti-galectin3 (1:1000, eBioscience, Affymetrix, Santa Clara, USA), mouse anti-α-tubulin (1:5000, Sigma), mouse anti-TRRAP (1;1000, Euromedex, Souffelweyersheim, France), rabbit anti-Sp1 (1:1000, Millipore), mouse anti-RB3/STMN4 (1:1000, Santa Cruz), rabbit anti-STMN3 (1:1000, Proteintech), mouse anti-GAPDH (1:1000, Sigma), mouse anti-β-actin (1:5000, Sigma), mouse anti-GFP (1:400, Santa Cruz), and rabbit anti-Oct4 (1:1000, Cell Signaling).

## Dendritic tree analysis

The tissues were fixed in 4% paraformaldehyde (PFA) and embedded in 4% low-melting agarose. 200 µm sagittal sections were obtained using a vibrating microtome HM 650 V (Thermo Scientific Microm) and mounted in slides. Imaging was performed using a Zeiss LSM 710 Confocal three microscope (Zeiss) and the Sholl analysis (Kroner et al., 2014; Sholl, 1953) achieved using the Fiji plugins Simple Neurite Tracing (Longair et al., 2011).

## Transmission electron microscopy

Mice were sacrificed using $CO_2$ and perfused intracardially with cold fixative (3% glutaraldehyde, 1% paraformaldehyde, 0.5% acrolein, 4% sucrose, 0.05 M $CaCl_2$ in 0.1 M cacodylate buffer, pH 7.3). The cerebellum was isolated and postfixed for a minimum of 1 day. For a secondary fixation, the samples were incubated in 2% $OsO_4$/1% potassium ferrocyanide in 0.1 M cacodylate buffer for 3 hr at 4°C in the dark, followed by dehydration in an ascending water/acetone series – then embedded in epoxy resin 'Epon' (glycid ether 100, SERVA, Heidelberg, Germany). The resin was allowed to polymerize for 2 days at 60°C in flat embedding molds. Ultrathin sections (50 nm) were produced using an ultra-microtome (Reichert Ultracut S; Leica, Wetzlar, Germany) and electron micrographs taken on a JEM 1400 electron microscope (JEOL, Akishima, Japan), using an accelerating voltage of 80 kV coupled with Orius SC 1000 CCD-camera (GATAN, Pleasanton, USA).

## Transcriptomics

The total RNA was isolated from tissues or cultured aNSCs using an RNeasy Lipid Tissue Mini Kit (Qiagen, Venlo, The Netherlands) and an RNAeasy Mini Kit (Qiagen) respectively, per manufacturer's instructions. The sequencing of RNA samples was done using Illumina's next-generation sequencing methodology (Bentley et al., 2008) – the quality check and quantification of the total RNA were

completed using the Agilent Bioanalyzer 2100 in combination with the RNA 6000 nano kit (Agilent Technologies). For library preparation 3 µg of tissue total RNA or 800 ng of aNSC total RNA were introduced to the TruSeq Stranded mRNA Kit (Illumina, San Diego, USA), per manufacturer's description. The quantification and quality check of the libraries were conducted using the Agilent Bioanalyzer 2100 in combination with the DNA 7500 kit. For sequencing of tissues, pools of four libraries were compiled and each pool was loaded on one lane of a HiSeq2500 machine running in 51cycle/single-end/high-output mode. For the sequencing of aNSCs, all libraries were pooled and loaded on three lanes of a HiSeq2500 machine running in 51 cycle/single-end/high-output mode. The sequence information was extracted in FastQ format using Illumina's bcl2FastQ v1.8.4. The sequencing resulted in around 55mio and 37mio reads per sample for tissues and aNSCs, respectively.

The reads of all samples were mapped to the mouse reference genome (GRCm38) with the Ensembl genome annotation (Release Ensembl 92) using STAR (version 2.5.4b; parameters: `–alignIntronMax` 100000 `–outSJfilterReads` Unique `–outSAMmultNmax` 1 `–outFilterMismatchNoverLmax` 0.04) (*Dobin et al., 2013*). Reads mapped uniquely to one genomic position were assigned to the gene annotated at this position with FeatureCounts (version 1.5.0; meta-feature mode, stranded mode '2', Ensembl 92) (*Liao et al., 2014*). A quality assessment of the raw input data, the read mapping and assignment steps, was performed using MultiQC (version 1.6) (*Ewels et al., 2016*), with the respective results provided in *Supplementary file 6*.

Read counts per gene were subjected to the R package DESeq2 (version 1.20.0) (*Love et al., 2014*), to test for differential expressions in pairwise comparisons as follows: Cultured aNSCs: five mutants contrasted to five controls; Cortex: four mutants contrasted to four controls; Striatum: four mutants contrasted to four controls. For each gene and comparison, the p-value was calculated using the Wald significance test. The resulting p-values were adjusted for multiple testing with Benjamini and Hochberg correction. Genes with an adjusted p<0.05 (false discovery rate, FDR) are considered differentially expressed. The log2 fold changes (LFC) were shrunk with lfcShrink from the DESeq2 package, to control for a variance of LFC estimates for genes with low read counts. The overlaps of all three pairwise DEG lists were calculated and visualized using the R package VennDiagram (version 1.6.20).

GO and KEGG pathway enrichment analyses were performed by supplying the gene lists of DEG overlaps into the database for annotation, visualization, and integrated discovery (DAVID) programs (*Huang et al., 2009a*; *Huang et al., 2009b*). TFBS enrichment analysis was performed by supplying the different lists of DEGs into the gene set enrichment analysis (GSEA) database (*Subramanian et al., 2005*; *Mootha et al., 2003*). A list of the Sp1 targets was extracted from the Harmonizome database (*Rouillard et al., 2016*) and compared with the RNA-seq data sets. The lists of Sp1 targets affected by the Trrap deletion was then analyzed using the Ingenuity Pathway Analysis (IPA) program (Qiagen).

## Sample preparation for MS proteomics

First, homogenates of the cortex tissues were prepared using the bead-beating device (24 tissue homogenizer) from Precellys (Montigny-le-Bretonneux, France). Frozen tissue was transferred on ice to bead-beating tubes (Precellys CKMix, 0.5 ml) containing ice-cold PBS with Protease and a Phosphatase Inhibitor cocktail (Roche) and beaten for 2 cycles of 20 s at 6000 rpm, with a 30 s break at 4°C. Homogenates were prepared at an estimated protein concentration of 10 µg/µl; based on 5% protein content of fresh brain tissues by weight. A volume of homogenate corresponding to approximately 500 µg protein was transferred to 1.5 ml Eppendorf tubes and taken for lysis. Lysis was carried out by resuspension of the homogenate in lysis buffer (final concentration 4% SDS, 0.1 M HEPES [pH 8], 50 mM DTT) to a final protein concentration of 1 µg/µl, followed by a sonication in a Bioruptor (Diagenode, Seraing, Belgium) (10 cycles, 1 min ON/30 s OFF, 20°C). The samples were heated (95°C, 10 min), and sonication steps repeated. The lysates were clarified by brief centrifugation, incubated with iodacetamide, (15 mM) at RT, in the dark. Each sample was treated with four volumes ice-cold acetone to precipitate the proteins (overnight, −20°C). The samples were centrifuged at 20,800 g (30 min, 4°C). The supernatant was removed and the pellets washed twice with 400 µl of ice-cold 80% acetone/20% water. The pellets were air-dried before dissolving in a digestion buffer (3 M urea in 0.1 M HEPES, pH 8) at 1 µg/µl. A 1:100 w/w amount of LysC (Wako, Richmond, USA; sequencing grade) was added to each sample before incubation (4 hr, 37°C, 1000 rpm).

The samples were diluted 1:1 with milliQ water and incubated with a 1:100 w/w amount of trypsin (Promega, Madison, USA; sequencing grade) (overnight, 37°C, 650 rpm). The digests were acidified with 10% trifluoroacetic acid and desalted with Waters Oasis HLB µElution Plate 30 µm (Waters, Milford, USA) in the presence of a slow vacuum, to manufacturer's instructions. The eluates were dried down with the speed vacuum centrifuge. Peptide labeling with TMT and subsequent high pH fractionation and LC-MS were conducted as detailed previously (*Buczak et al., 2018*). Briefly, the peptide samples obtained from the digestion were labeled with TMT-10plex isobaric mass tags (Thermo Fischer Scientific) per manufacturer's instructions. Equal amounts of the labeled peptides from the 10 samples (five replicates each condition) were mixed, desalted, and pre-fractionated into 16 fractions using high pH reverse phase fractionation on an Agilent Infinity 1260 HPLC, then each fraction was measured individually by nano-LC-MS on an Orbitrap Fusion Lumos employing SPS-MS3 data acquisition (Thermo Fischer Scientific). Subsequently, the fraction data were searched together in Mascot 2.5.1 (Matrix Science, Boston, USA) using Proteome Discoverer 2.0 (Thermo Fischer Scientific) against the Swissprot *Mus musculus* database (2016; 16,756 entries) and a list of common contaminants. Reporter ion intensity values for the PSMs were exported and processed using in-house written R scripts to remove common contaminants and decoy hits. Only PSMs having reporter ion intensities above $1 \times 10^3$ in all the relevant TMT channels were retained for a quantitative analysis, as described in *Buczak et al., 2018*. Briefly, the reporter ion (TMT) intensities were $\log_2$-transformed and normalized. Peptide-level data were summarized into their respective protein groups by taking the median value. For differential protein expression, the five replicates of the two conditions respectively within the TMT10-plex were taken together. Protein ratios were calculated for all protein groups quantified with at least two peptides. To compare DEP in the cortex obtained by RNA-seq to protein DEP (differentially expressed proteins) obtained by mass spectrometry, Ensembl gene IDs were mapped to Uniprot IDs with the R packages AnnotationDbi (1.42.1) and org.Mm.eg.db (3.6.0), while only genes/proteins present in both analyses were considered. When for a single Uniprot ID multiple Ensembl IDs are known, the proteomics measurement is duplicated and all different transcriptomics results assigned to this entry.

## aNSC cell culture

The SVZ of 2–4 months old mice were isolated, minced, and digested with DMEM/F-12 medium supplemented with 20 U/ml papain, 240 µg/ml cysteine, and 400 µg/ml DNAse I type IV. After 1 hr, the digestion was stopped by ovomucoid trypsin inhibitor. The homogenized aNSCs were then cultured in suspension medium (DMEM/F-12 medium supplemented with 1× B-27 Supplements, 1× penicillin–streptomycin, 20 ng/ml EGF, 20 ng/ml bFGF). To induce Trrap deletion, aNSCs were treated with 1 µM 4-hydroxytamoxifen (4-OHT) for 3 days, followed by incubation in fresh medium for another 2 days.

## Transfection, Sp1 knockdown, and luciferase assay

$2 \times 10^5$ aNSCs were plated in 50 µg/ml PLL and 10 µg/ml laminin pre-coated 24-well plates in Neurobasal Medium (NEM) supplemented with 1× B-27 Supplements, 2 mM L-glutamine, 1× N-2 Supplement, 1× penicillin–streptomycin, 20 ng/ml EGF, and 20 ng/ml bFGF. The transfection was performed after overnight culture using Lipofectamine 2000. For luciferase assay of Sp1 activity, the vector −111 hr TF m3 was used as Sp1 reporter plasmid – gifted by Nigel Mackman (Addgene plasmid # 15450; http://n2t.net/addgene:15450; RRID:Addgene_15450). Guntram Suske gifted the vector pN3-Sp1FL used to overexpress Sp1 (Addgene plasmid # 24543; http://n2t.net/addgene:24543; RRID:Addgene_24543). 24 hr later, transfection cells were collected to measure the luciferase activity using a Dual-Glo Luciferase Assay System (Promega), per manufacturer's instructions. For Sp1 knockdown, aNSC in adherent conditions was supplemented with 30 nM siRNA against Sp1 mixed with RNAiMAX reagent (Thermo Fischer Scientific). After 48 hr, transfection cells were collected for immunoblot analysis.

## Chromatin preparation for ChIP and ChIP-seq

$2 \times 10^6$ aNSCs were cross-linked by adding formaldehyde 1% for 10 min at room temperature, quenched with 0.125 M glycine for 5 min at room temperature, then washed three times in phosphate-buffered saline (PBS) before freezing. Pellets were suspended in 0.25 ml SDS lysis buffer (50

mM HEPES-KOH, 140 mM NaCl, 1 mM EDTA, 0.1% Triton X-100, 0.1% sodium deoxycholate, 1% SDS, 10 mM NaB, and protease inhibitors), incubated on a rotator for 30 min at 4°C, sonicated for 20 min at 4°C, then centrifuged at 14,000 rpm for 10 min at 4°C. Supernatants were diluted 10-fold with a ChIP dilution buffer (1% Triton X-100, 2 mM EDTA, 20 mM Tris-HCl, 150 mM NaCl, 10 μM NaB, and protease inhibitors) (25 μl retained as input) and incubated overnight in gentle rotation at 4°C with 4 μg of antibody. The following antibodies were used: rabbit anti-SP1 (Millipore), rabbit anti-acetyl-Histone 3 (Millipore), mouse anti-TRRAP (Euromedex), rabbit anti-acetyl-Histone 4 (Millipore), rabbit anti-H3K4me2 (Abcam), rabbit anti-H3 (Abcam), rabbit anti-H4 (Abcam), rabbit anti-Oct4 (Cell Signaling), and rabbit anti-IgG (Sigma). After that, 40 μl of preblocked protein-G-conjugated magnetic beads (DYNAL, Thermo Fischer) were added and incubated for 2 hr in a rotator at 4°C. The immunoprecipitated complexes were washed three times in low-salt wash buffer (0.1% SDS,1% Triton X-100, 2 mM EDTA, 20 mM Tris-HCl, 150 mM NaCl), once in high-salt wash buffer (0.1% SDS, 1% Triton X-100, 2 mM EDTA, 20 mM Tris-HCl, 500 mM NaCl) and once in TE buffer. The complexes were eluted by adding 0.2 ml of Elution buffer (TE 1×, 1% SDS, 150 mM NaCl, 5 mM DTT) for 30 min in rotation at room temperature. The de-cross-linking was performed overnight at 65°C. The de-cross-linked DNA was purified using a QiaQuick PCR Purification Kit (Qiagen) according to the manufacturer's instruction.

## ChIP-seq

For the library preparation, approximately 10 ng of purified ChIP DNA was end-repaired, dA-tailed, and adaptor-ligated using the TruSeq ChIP Sample Preparation Kit (illumina), to manufacturer's instructions. The size of the library was checked using Fragment Analyzer (Agilent) and the library sequenced on the NextSeq500 platform (illumina). The Fastq files quality check was performed with FastQC (v0.11.5). Fastq files mapping to mm9 genome was performed by using Bowtie (v1.1.2) with `–best –strata` -m one parameters. Duplicate reads were removed using a custom script. For peak calling, macs14 (v1.4.2) was used with `–nolambda` parameter and two different p-value cutoffs (1e-3 for histone modifications and 1e-5 for SP1). Other downstream analyses were done using R (v3.4.4). For a RPM (Read Per Million) calculation, the peaks were merged using the Peakreference function (TCseq_1.2.0 package). The merged peaks were used as the reference for the calculation of RPM for each sample by using a custom script. 10% or 30% of the most depleted regions in mutant versus control samples for histone modifications and Sp1 respectively were used as cutoff for defining differentially regulated regions. Differentially regulated regions were assigned to the nearest gene (ENSEMBL annotation), where the distance of the region was less than ±5 Kb to the TSS (Transcription Start Site).

ChIP qRT-PCR was performed using the Platinum SYBR Green qPCR SuperMix-UDG (Thermo Fischer Scientific) and a LightCycler 480 Real-Time PCR System (Roche). All experiment values were subtracted by those obtained with a rabbit nonimmune serum (IgG) and divided by input, as indicated in the literature (*Neri et al., 2012*). The following primers were used for amplification: STMN3: Fwd, 5'-CTTGCTACTGCATCAGGCGA-3'; Rev, 5'-AGCCTAGGGGATCATGGGAC-3'; STMN4: Fwd, 5'-TCGCTTTGGAAACCGGACTG-3'; Rev, 5'-TTTGTTTAAAACCCCCGCCC-3'.

| siRNA | Sequence (5'→ 3') |
|---|---|
| siScramble | UGGUUUACAUGUCGACUAA |
| siTrrap-1 | CAAAAGUAGUGAACCGCUA |
| siTrrap-2 | CCUACAUUGUGGAGCGGUU |
| siTrrap-3 | GCCAACUGUCAGACCGUAA |
| siTrrap-4 | CGUACCUGGUCAUGAACGA |
| siSp1 | GGAUGGUUCUGGUCAAAUAtt |

## Acknowledgements

We thank D Stefanova, H Ivashenko, S Ortega, S Tsukamoto, M Oi, C Murakami, and C Meisezahl for their technical support and assistance. We also thank P Elsner for his excellent assistance in

maintenance of the animal colonies and K Buder for her help using TEM. We are grateful to Dr K-H Gührs for his critical reading of the manuscript. We are grateful to the FLI Core Facilities DNA Sequencing, Life Science Computing, Proteomics, Histology and Imaging for their technical support. We thank members of the Wang Laboratory for their helpful and critical discussions. AT was supported by a Postdoc Fellowship of the Leibniz Institute for Age Research (FLI) and DL was a recipient of a PhD studentship from the Leibniz Graduate School on Aging (LGSA). Z-QW is supported in part by the Deutschen Forschungsgemeinschaft (DFG), Germany.

## Additional information

### Funding

| Funder | Grant reference number | Author |
| --- | --- | --- |
| Leibniz Association | Postdoc Fellowship | Alicia Tapias |
| Leibniz Association | PhD studentship | David Lázaro |
| Leibniz Association | Open Access Fund | Zhao-Qi Wang |

The funders had no role in study design, data collection and interpretation, or the decision to submit the work for publication.

### Author contributions

Alicia Tapias, Conceptualization, Data curation, Formal analysis, Investigation, Methodology, Writing - original draft, Writing - review and editing; David Lázaro, Conceptualization, Data curation, Formal analysis, Investigation, Writing - original draft; Bo-Kun Yin, Data curation, Formal analysis, Investigation; Seyed Mohammad Mahdi Rasa, Anna Krepelova, Erika Kelmer Sacramento, Formal analysis, Investigation; Paulius Grigaravicius, Formal analysis, Investigation, Methodology; Philipp Koch, Formal analysis; Joanna Kirkpatrick, Data curation, Formal analysis, Investigation, Methodology; Alessandro Ori, Conceptualization, Investigation, Methodology; Francesco Neri, Conceptualization, Data curation, Formal analysis; Zhao-Qi Wang, Conceptualization, Data curation, Supervision, Funding acquisition, Investigation, Writing - original draft, Project administration, Writing - review and editing

### Author ORCIDs

Seyed Mohammad Mahdi Rasa [iD] http://orcid.org/0000-0001-6850-8909
Philipp Koch [iD] http://orcid.org/0000-0003-2825-7943
Alessandro Ori [iD] http://orcid.org/0000-0002-3046-0871
Zhao-Qi Wang [iD] https://orcid.org/0000-0002-8336-3485

### Ethics

Animal experimentation: Animal experiments were conducted according to German animal welfare legislation, and the protocol is approved by Thüringen Landesamt für Verbraucherschutz (TLV) (03-042/16), Germany.

### Decision letter and Author response

Decision letter https://doi.org/10.7554/eLife.61531.sa1
Author response https://doi.org/10.7554/eLife.61531.sa2

## Additional files

### Supplementary files

• Source data 1. SP1-regulated molecular pathways. (**A**) Top30 nervous system processes targets of Sp1. DEGs in all three RNA-seq data sets were compared with the list of the Sp1 targets from the Harmonizome database (*Rouillard et al., 2016*) and the resulting list was analyzed using IPA to find the disease process associated with the DEGs (cutoff, $p < 0.05$). (**B**) Top30 differentially expressed Sp1 targets. DEGs in the RNA-seq data sets were compared with the list of Sp1 targets from the

Harmonizome database and the Top30 DEGs (cutoff, p<0.05) are indicated. (**C**) Top30 protein changes of Sp1 targets. Proteins from the forebrain, whose expression changed after the Trrap deletion and correlated with the changes in RNA-seq, were compared with the list of Sp1 targets obtained from the Harmonizone database. The Top30 results based on the *q*-value are summarized.

• Supplementary file 1. The list of up- and downregulated genes (adjusted p-value <0.05) in different data sets. The list includes the DEGs in Trrap-FBΔ cortices (**A**), Trrap-FBΔ striata (**B**), and Trrap-deleted aNSCs (**C**). The list also includes comparisons between Trrap-FBΔ cortices and striata (**D**) and Trrap-FBΔ cortices, striata, and Trrap-deleted aNSCs (**E**). Moreover, it includes the GO (**F**) and KEGG (**G**) terms obtained from the list in (**D**), statistical information, and the list of genes in each group.

• Supplementary file 2. The list of protein changes in Trrap-FBΔ cortices. (**A**) The list of protein changes after Trrap deletion. (**B**) The comparison between protein changes (proteomics, *q* < 0.1) and mRNA changes (transcriptomics, adjusted p-value <0.05).

• Supplementary file 3. The results of TFBS enrichment analysis in different data sets. The list includes the results from the TFBS enrichment analysis using the following lists as a template: (**A**) DEGs in D. (**B**) First 2940 DEGs from the list in A sorted by adjusted p-value. (**C**) DEGs 2941 to 5090 from the list in A sorted by adjusted p-value. (**D**) First 2940 DEGs from the list in B sorted by adjusted p-value. (**E**) DEGs 2941 to 4741 from the list in B sorted by adjusted p-value. (**F**) DEGs in E.

• Supplementary file 4. Changes in Sp1 targets in different data sets. (**A**) A list of known Sp1 targets was obtained from the Harmonizome database (*Dubey et al., 2015*). The common gene names were transformed to Ensembl gene IDs using the online conversion tool from the DAVID database. (**B**) A comparison between the DEGs in Suppl. File 1A and the known Sp1 targets listed in (**A**). (**C**) A comparison between the DEGs in Suppl. File 1B and the known Sp1 targets listed in (**A**). (**D**) A comparison between the DEGs in Suppl. File 1D and the known Sp1 targets listed in (**A**). (**E**) The GO terms obtained from the list in (**D**), statistical information, and the list of genes in each group. (**F**) The KEGG terms obtained from the list in (**D**), statistical information, and the list of genes in each group. (**G**) The list includes the results of the Sp1 ChIP-seq in aNSCs. 30% of the most depleted regions in Trrap-D versus control aNSCs for Sp1 were used as cutoff for defining differentially regulated regions. (**H**) A comparison between the ChIP-seq results in (**G**) and the known Sp1 targets listed in (**A**). (**I**) The overlaps between the ChIP-seq results in (**G**) and the DEGs in Suppl. File 1D. (**J**) A comparison between the DEGs in (**I**) and the known Sp1 targets listed in (**A**).

• Supplementary file 5. Changes in acetylation after Trrap deletion. (**A**) The list includes the results of AcH3 ChIP-seq of aNSCs. The 10% most depleted regions in Trrap-Δ versus control aNSCs for AcH3 are summarized. (**B**) The list includes the results of the AcH4 ChIP-seq of aNSCs. The 10% most depleted regions in Trrap-Δ versus control aNSCs for AcH4 are summarized. (**C**) The overlaps between the genes mapped in (**A**) and (**B**). (**D**) The overlaps between the DEGs in D and the genes mapped in (**A**). (**E**) The overlaps between the DEGs in D and the genes mapped in (**B**). (**F**) The overlaps between the DEGs in D and the genes mapped in (**C**). (**G**) A combined list from (**D**) and (**E**) was created and compared with the list of Sp1 targets in *Supplementary file 4A*. The list contains genes where the acetylation of H3 or H4 was decreased, whose expression was altered after Trrap deletion and which are reported Sp1 targets.

• Supplementary file 6. Quality assessment of RNA-seq raw input data. The table provides the results of the read mapping and assignment steps performed using MultiQC (version 1.6) (*Ewels et al., 2016*).

• Transparent reporting form

## Data availability

The data discussed in this publication have been deposited in NCBI's Gene Expression Omnibus and are accessible through the GEO Series accession numbers GSE131213 (RNA-seq aNSCs), GSE131283 (RNA-seq brain tissues) and GSE131028 (ChIP-seq aNSCs). The mass spectrometry proteomics data have been deposited to the ProteomeXchange Consortium (http://proteomecentral.proteomexchange.org) via the PRIDE partner repository, with the dataset identifier PXD013730.

The following datasets were generated:

| Author(s) | Year | Dataset title | Dataset URL | Database and Identifier |
|-----------|------|---------------|-------------|-------------------------|
| Kirkpatrick J, Ori A, Wang ZQ | 2021 | The mass spectrometry proteomics of different brain areas of Trrap conditional knockout *Mus musculus*. | https://www.ebi.ac.uk/pride/archive/projects/PXD013730 | PRIDE, PXD013730 |
| Groth M, Koch P, Pellón DL, Wang ZQ | 2021 | RNA-seq of murine primary adult stem cells of Trrap inducible knockout *Mus musculus* with and without 4-OHT treatment | https://www.ncbi.nlm.nih.gov/geo/query/acc.cgi?acc=GSE131213 | NCBI Gene Expression Omnibus, GSE131213 |
| Groth M, Koch P, Pellón DL, Soler AT, Wang ZQ | 2021 | RNA-seq of different brain tissue areas of Trrap conditional knockout *Mus musculus* | https://www.ncbi.nlm.nih.gov/geo/query/acc.cgi?acc=GSE131283 | NCBI Gene Expression Omnibus, GSE131283 |
| Krepelova A, Rasa SMM, Neri F, Wang ZQ | 2021 | ChIP-seq of adult neuro-stem cells of Trrap inducible knockout *Mus musculus* with and without 4-OHT treatment. | https://www.ncbi.nlm.nih.gov/geo/query/acc.cgi?acc=GSE131028 | NCBI Gene Expression Omnibus, GSE131028 |

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
