## [Decision Letter]

**Acceptance summary:**

This manuscript identifies a novel mechanism linking loss of Transformation/transcription domain-associated protein (TRRAP) to neurodegeneration and motor deficits in a mouse model. Overall, the compelling results highlight a regulatory pathway by which TRRAP, the transcription factor SP1, and histone acetylation interact to control expression of key microtubule associated genes, resulting in destabilized microtubule dynamics when TRRAP is deleted. Given the ties between TRRAP mutation and several human neuropathies, these findings have implications for uncovering novel disease mechanisms.

**Decision letter after peer review:**

Thank you for submitting your article "TRRAP-HAT modulates microtubule dynamics via SP1 signaling in brain homeostasis and neurodegeneration" for consideration by *eLife*. Your article has been reviewed by three peer reviewers, one of whom is a member of our Board of Reviewing Editors, and the evaluation has been overseen by Huda Zoghbi as the Senior Editor. The reviewers have opted to remain anonymous.

The reviewers have discussed the reviews with one another and the Reviewing Editor has drafted this decision to help you prepare a revised submission.

Summary:

This manuscript from Tapias, Larazo, et al., examines effects of neuronal deletion of the transformation/transcription domain associated protein (Trrap) gene. The authors show that Trrap deletion in Purkinje neurons of the cerebellum causes neurite retraction, progressive cell loss, and age-related motor control deficits. Similarly, deletion of Trrap in the cortex and striatum (using the forebrain-selective Camk2a promoter) results in widespread transcriptional dysregulation, specifically at genes containing SP1 transcription factor motifs. Consistent with SP1-mediated deficits in gene expression, Trrap deletion decreased SP1 activity at a luciferase reporter, and decreased SP1 binding and histone acetylation at Trrap target genes Stmn3 and Stmn4. Similar gene expression changes were observed after conditional Trrap deletion in neural stem cells. Finally, the authors report that overexpression of Trrap target genes Stmn3 and Stmn4 rescues effects of Trrap knockdown on neurite branching and length in primary cultured neurons.

Overall, the results identify a novel mechanism by which Trrap regulates neuronal function, and might contribute to neuronal development and potentially also neurodegeneration. Additionally, these results may have relevance to Trrap mutations in human patients.

While the results identify a potentially novel link between Trrap, SP1, and microtubule dynamics that support neruonal growth and function, the reviewers had several concerns with the current manuscript. First, the evidence that SP1 and histone acetylation mediates the effects of Trrap deletion is not sufficient to fully support the claims of the manuscript. Second, while the examination of different cell types is helpful and increases the ability to generalize effects of Trrap deletion, there are significant concerns about the timing of Trrap deletion in these models, and several of the results are not consistent with a neurodegenerative phenotype. We have suggested the following revisions to improve the manuscript.

Essential revisions:

1) The data presented in Figure 3G are intriguing, and are the main data in the paper to support a direct mechanistic link between Trrap and SP1 function. However, these results are based on a relatively underpowered experiment (n=3), with large error. The manuscript would be improved if this could be strengthened or bolstered by additional evidence. For example, the manuscript already contains H3ac and H4ac ChIP-seq data, which is used to infer effects mediated by SP1 at Trrap target genes. The authors might confirm this relationship by looking at the distribution of histone acetylation across SP1 sites in the genome (as well as specifically at Trrap DEGs). Similarly, it would be useful to show that Trrap is normally found at SP1 motifs for these genes in neurons (e.g. with ChIP). Likewise, although the ChIP data suggest that SP1 regulates STMN3 and 4 expression, this is not directly demonstrated. Does SP1 depletion alone influence their expression?

2) The authors argue that neurodegeneration linked with Trrap deletion is the result of altered transcription of SP1 targets through impaired SP1 activity caused by reduced acetylation. However, the overlap between hypoacetylated loci and SP1-regulated genes on a genome-wide scale was pretty low (subsection “Trrap-HAT mediates Sp1 transcriptional control of microtubule dynamic genes”, 11 genes), indicating that other factors may be at play. The authors validate their assertion by showing that Sp1 and AcH4 are reduced by Trrap deletion on target promoters (Figure 4E and F), but this study is lacking proper controls to show that these changes are selective for acetylation. Given that the authors argue that the effect is mediated by HAT activity (which is not measured but inferred based on known Trrap functions), it is important to exclude alternative explanations, such as non-specific effects of Trrap deletion on nucleosome density or other post-translational modifications. As such, studies in Figure 4D should be repeated for total H4 and also for another PTM that isn't directly regulated by Trrap. In addition, to validate the relevance of SP1 as a regulator of genes affected by Trrap function, an additional control would be beneficial – specifically, the authors could conduct ChIP for a TF that was not associated with genes affected by Trrap deletion in Figure 3F to show that changes in Figure 4F are specific for SP1 and acetylation.

3) The authors link SP1 with impaired microtubule function. They show microtubule-related terms in ontology analysis of differentially regulated genes that overlap in the striatum and the cortex, but not specifically for DEGs that contain SP1 binding sites. Although they do identify STMN3 and 4 as targets, it would be useful to see if these categories come out for SP1 regulated genes as a group. As authors point out, SP1 tends to have wide spread functions that influence many pathways, so it is important to exclude the possibility that microtubule-related functions affected by Trrap may be small compared to other pathways that are influenced more dramatically. Even if they are small, this finding is still of interest, but the conclusions need to be dampened.

4) While the manuscript claims that TRRAP-HAT modulates microtubule dynamics via SP1 signaling in brain homeostasis and neurodegeneration, the evidence for these assertions is relatively weak. The signaling pathway the authors found is based on the RNA-seq and ChIP-seq in the forebrain at P10 when postnatal brain development and maturation is still going on. Also, the role of TRRAP-HAT-SP1-STMN3/4 signaling in the neurodegeneration context is not validated since the cortical primary culture at DIV6-12 is also at developmental stage rather than fully matured stage. Therefore, while the manuscript begins with Purkinje neuronal degeneration by Trrap deletion, data in Figure 3 shows that the signaling pathway is more likely to be involved in the neural development. Similarly, the authors did not carefully confirm the effect of Trrap deletion on Purkinje neuronal development. In the PCP2-Cre line the authors used in this study, Cre expression starts at P6 (Zhang et al., 2004). Considering the cerebellum and Purkinje neuronal development is completed around at ~P20, it is difficult to exclude the possibility TRRAP-HAT-SP1-STMN3/4 signaling affect the Purkinje neuronal development rather than neuronal survival. Indeed, previous study showed that STMN3 is crucial for the formation and development of the Purkinje cell dendritic arbor (Poulain et al., 2008). These issues should be addressed in the manuscript, and if the authors cannot rule out neurodvelopmental effects, the specific claims regarding neurodegeneration should be removed.

5) The primary interest appears to be in investigating the role of Trrap in cerebellar neurodegeneration, and as such, they quantify phenotypes that are relevant to cerebellar function. However, mechanistic studies are conducted in the striatum and cortex. The argument for switching mouse models and brain regions was the scarcity of Purkinje cells in the Trrap-PC model, so they switch to CamKIIa Cre and striatum and cortex for mechanism. A search of the literature and Allen Brain atlas shows CamK2a expression in cerebellar cells, including Purkinje cells. Since there was no cell sorting performed to select for CamK2a cells in any of the brain regions examined, the experiments would presumably be able to be conducted in the cerebellum? Similarly, the authors use multiple cell types in cell culture studies, including cultured cortical neurons – why not use cultured cerebellar neurons?

6) Please clarify how the ChIP datasets in Figure 4E-F are compared between groups. Why is there no variability in the control group? For ChIP-qPCR data, it would be preferable to show the results as % of input, rather than binding enrichment. Additionally, while the sequences for ChIP-qPCR primers are provided, the authors should clarify where these primers target within the genome, and whether site contains a known SP1 motif.

7) The data showing siRNA-mediated Trrap knockdown (Figure 5—figure supplement 1A) is not convincing – summary data and statistical comparisons should be shown for manipulations as performed in the in vitro experiments. Similarly, it is not clear how transfection experiments would lead to robust knockdown. Typically, neuronal transfection experiments suffer from a very low efficiency, meaning that manipulations would only occur in a very small fraction of cells. If this is not the case here, data for transfection efficiency should be provided.

8) The Title of the manuscript should be revised to meet *eLife* guidelines. Specifically, the Title should avoid use of dashes, acronyms, or unfamiliar abbreviations (unless needed for scientific reasons). Please revise your Title with this advice in mind. Additionally, the Title and/or Abstract should provide a clear indication of the biological system under investigation (i.e., species name or broader taxonomic group, if appropriate). Please revise your Title and/or Abstract with this advice in mind.

[Editors' note: further revisions were suggested prior to acceptance, as described below.]

Thank you for submitting your article "HAT cofactor TRRAP modulates microtubule dynamics via SP1 signaling in neurodegeneration" for consideration by *eLife*. Your article has been reviewed by three peer reviewers, one of whom is a member of our Board of Reviewing Editors, and the evaluation has been overseen by Huda Zoghbi as the Senior Editor. The following individual involved in review of your submission has agreed to reveal their identity: Hongjun Song (Reviewer #3).

The reviewers have discussed the reviews with one another and the Reviewing Editor has drafted this decision to help you prepare a revised submission.

We would like to draw your attention to changes in our policy on revisions we have made in response to COVID-19 (https://elifesciences.org/articles/57162). Specifically, when editors judge that a submitted work as a whole belongs in *eLife* but that some conclusions require a modest amount of additional new data, as they do with your paper, we are asking that the manuscript be revised to either limit claims to those supported by data in hand, or to explicitly state that the relevant conclusions require additional supporting data.

Summary:

In original manuscript, the authors identified a novel mechanism by which Trrap regulates neuronal function, with implications for neurodegeneration. Specifically, they show that loss of TRRAP results in neurite retraction, neurodegeneration, and motor deficits. They link these changes with activity of the transcription factor SP1 and with two downstream targets, Stmn3 and Stmn4, which are implicated in microtubule dynamics. Overall, these data present a novel regulatory pathway in regulating microtubule dynamics and provide a basis for uncovering the role of specific epigenetic factors in this process. Given the ties between TRRAP and several human conditions affecting neural conditions, these findings have implications for uncovering novel disease mechanisms.

In this revised manuscript, the authors made significant efforts to address all the concerns raised by previous reviews with new experiment, data re-analyses, and clarification in the text. The manuscript is significantly improved, and the new results support the author's interpretation that Trrap deletion alters SP1 binding and expression of STMN3 and STMN4. Notably, overexpression of STNM3 can rescue certain neurite growth deficits caused by Trrap deletion, in agreement with their model. Likewise, many conclusions have been appropriately toned down to match the experimental results or in light of noted caveats. However, some specific revisions are still critical prior to publication of the manuscript (outlined below).

Essential revisions:

1) Replace all figures with higher resolution images – the current versions appear pixelated upon zooming in to see details. Vector images are preferred where possible.

2) In relation to comment 4: Conclusions regarding neurodegeneration are still too strong and the answer provided in response to the reviewers is not fully reflected in the discussion of the revised manuscript.

3) The authors still do not address the lack of variability in control groups in several figures, including:

-Figure 4B – from blot images in Figure 4A, there seems to be sample variability in controls, but the individual data points look exactly the same. The same applies to Figure 5 – figure supplement 1A.

-Individual data points are shown for some graphs, but not for others. Individual data points should be included for all graphs.

---

## [Author Response]

Essential revisions:1) The data presented in Figure 3G are intriguing, and are the main data in the paper to support a direct mechanistic link between Trrap and SP1 function. However, these results are based on a relatively underpowered experiment (n=3), with large error.

We have now newly performed the experiments 5 times. The new data replaced the old Figure 3G by revised Figure 3F.

The manuscript would be improved if this could be strengthened or bolstered by additional evidence. For example, the manuscript already contains H3ac and H4ac ChIP-seq data, which is used to infer effects mediated by SP1 at Trrap target genes. The authors might confirm this relationship by looking at the distribution of histone acetylation across SP1 sites in the genome (as well as specifically at Trrap DEGs). Similarly, it would be useful to show that Trrap is normally found at SP1 motifs for these genes in neurons (e.g. with ChIP).

We appreciate this important suggestion and have reanalyzed the ChIP-seq data. The new analyses are shown as Figure 4—figure supplement 1B-C. Moreover, our ChIP assay also showed Trrap binding at Sp1 motifs (Figure 4F-G), because these primers contain Sp1 motifs, we now show these in Figure 4—figure supplement 1G. These new data are also described in the text.

Likewise, although the ChIP data suggest that SP1 regulates STMN3 and 4 expression, this is not directly demonstrated. Does SP1 depletion alone influence their expression?

To address this point, we performed siRNA mediated SP1 knockdown and found a reduction of STMN3 and 4 expressions. Representative Western blots and quantifications are provided in Figure 4H.

2) The authors argue that neurodegeneration linked with Trrap deletion is the result of altered transcription of SP1 targets through impaired SP1 activity caused by reduced acetylation. However, the overlap between hypoacetylated loci and SP1-regulated genes on a genome-wide scale was pretty low (subsection “Trrap-HAT mediates Sp1 transcriptional control of microtubule dynamic genes”, 11 genes), indicating that other factors may be at play. The authors validate their assertion by showing that Sp1 and AcH4 are reduced by Trrap deletion on target promoters (Figure 4E and F), but this study is lacking proper controls to show that these changes are selective for acetylation. Given that the authors argue that the effect is mediated by HAT activity (which is not measured but inferred based on known Trrap functions), it is important to exclude alternative explanations, such as non-specific effects of Trrap deletion on nucleosome density or other post-translational modifications. As such, studies in Figure 4D should be repeated for total H4 and also for another PTM that isn't directly regulated by Trrap. In addition, to validate the relevance of SP1 as a regulator of genes affected by Trrap function, an additional control would be beneficial – specifically, the authors could conduct ChIP for a TF that was not associated with genes affected by Trrap deletion in Figure 3F to show that changes in Figure 4F are specific for SP1 and acetylation.

We thank the reviewers for their valuable suggestions. To address the comments, we performed all these experiments and presented them as Figure 4F-G. We used H3 and H4 for histone controls and H3K4me2 as a non-relevant PTM control. Oct4 is a transcription factor that is not under the control of SP1 (PMID:10592386) and is used as a control, as requested by the reviewers. Please note there was a large error bar in the Oct4 ChIP data because one pair of the sample gave a very high value (we noted in the figure legend). However, when we compared pairwise (paired t-test), the error bar in both control and mutant is low (data not shown). Nonetheless, both showed no statistical difference. In revised manuscript, we present these new data in the text and described them in the figure legend.

3) The authors link SP1 with impaired microtubule function. They show microtubule-related terms in ontology analysis of differentially regulated genes that overlap in the striatum and the cortex, but not specifically for DEGs that contain SP1 binding sites. Although they do identify STMN3 and 4 as targets, it would be useful to see if these categories come out for SP1 regulated genes as a group. As authors point out, SP1 tends to have wide spread functions that influence many pathways, so it is important to exclude the possibility that microtubule-related functions affected by Trrap may be small compared to other pathways that are influenced more dramatically. Even if they are small, this finding is still of interest, but the conclusions need to be dampened.

Our original Figure 3E and Table 3 showed the overall changes of molecular pathways. Following this comment, we have now provided the Top50 of the SP1-containing DEGs in Figure 3—figure supplement 2G. As one can see, again, the microtubule dynamics is prominent in the list.

4) While the manuscript claims that TRRAP-HAT modulates microtubule dynamics via SP1 signaling in brain homeostasis and neurodegeneration, the evidence for these assertions is relatively weak. The signaling pathway the authors found is based on the RNA-seq and ChIP-seq in the forebrain at P10 when postnatal brain development and maturation is still going on. Also, the role of TRRAP-HAT-SP1-STMN3/4 signaling in the neurodegeneration context is not validated since the cortical primary culture at DIV6-12 is also at developmental stage rather than fully matured stage. Therefore, while the manuscript begins with Purkinje neuronal degeneration by Trrap deletion, data in Figure 3 shows that the signaling pathway is more likely to be involved in the neural development. Similarly, the authors did not carefully confirm the effect of Trrap deletion on Purkinje neuronal development. In the PCP2-Cre line the authors used in this study, Cre expression starts at P6 (Zhang et al., 2004). Considering the cerebellum and Purkinje neuronal development is completed around at ~P20, it is difficult to exclude the possibility TRRAP-HAT-SP1-STMN3/4 signaling affect the Purkinje neuronal development rather than neuronal survival. Indeed, previous study showed that STMN3 is crucial for the formation and development of the Purkinje cell dendritic arbor (Poulain et al., 2008). These issues should be addressed in the manuscript, and if the authors cannot rule out neurodvelopmental effects, the specific claims regarding neurodegeneration should be removed.

We are aware of the limitation of the model that was used. As stated in the text, the problem for Purkinje cell specific knockout is the scarcity of Purkinje cells (PC) in our PCdel models. Therefore, we switched to forebrain deleted model to analyze the pathway controlled by Trrap for the feasibility. To gain insight we performed integrated omics (RNA-seq, proteomics and ChIP-seq) and found a striking commonality of molecular pathways governed by TRRAP-HAT in different neural cell types. We provided detailed analyses on these omic datasets. Then, we confirmed STMNs using primary neurons and PC models (Figure 4C-D). However, given the normal PC development in the PCD model, i.e., a normal PC density and behavior at young age (Figure 1A-B), Trrap-Sp1-STMNs does not likely play a prominent role in early PC development at least not in the Pcp2-Cre model. Moreover, in our animal models the deletion occurs after the neuron differentiation, despite not fully matured, pointing out that at the early stages the Trrap deleted brain looks normal (Figure 1C, Figure 2C (Purkinje-Cre related) and Figure 3A (CamkII-Cre related)). Having said this, we have no direct evidence, which would allow us to rule out the possibility that Trrap-Sp1-STMNs affects the general neuronal development or survival. Indeed, as the reviewers pointed out, STMN3/4 are also important in the maintenance of neuron dendrites, which had been discussed in the original text. Thus, we have carefully rephrased our statement, in the revised manuscript.

It is noteworthy that we have many PCdel models running in the lab, in which many life essential genes were deleted, such as ATR, MRE11 and NBS1 (null mutation of these genes causes cell and embryonic lethality). None of these PCD models show the neurodegeneration phenotype (even after 20 months) in contrast to the Trrap-PCD model. These observations thus highlight the specificity of the HAT-TRRAP-Sp1-STMN axis in preventing PC degeneration.

5) The primary interest appears to be in investigating the role of Trrap in cerebellar neurodegeneration, and as such, they quantify phenotypes that are relevant to cerebellar function. However, mechanistic studies are conducted in the striatum and cortex. The argument for switching mouse models and brain regions was the scarcity of Purkinje cells in the Trrap-PC model, so they switch to CamKIIa Cre and striatum and cortex for mechanism. A search of the literature and Allen Brain atlas shows CamK2a expression in cerebellar cells, including Purkinje cells. Since there was no cell sorting performed to select for CamK2a cells in any of the brain regions examined, the experiments would presumably be able to be conducted in the cerebellum?

Our lab works on CamKII-Cre mice extensively and observed using Confetti reporter that CamKII-Cre can deleted only minor subpopulations of cells in cerebella. Thus, an extensive cerebellar analysis of Trrap in the CamkII-Cre model would be very complicated due to two reasons: we would have to first map exactly the cells deleted and second exclude the possible influence of the forebrain excitatory neurons (extensively expressing CamKII). In our Trrap-FBD model it is not even possible since these mutant mice die around weaning. Thus, CamKII-Cre is not a suitable model to study cerebella. Nevertheless, this model was very useful to gain insight into the mechanistic understanding on how Trrap regulates neurological processes.

Similarly, the authors use multiple cell types in cell culture studies, including cultured cortical neurons – why not use cultured cerebellar neurons?

Indeed, we tried to establish cerebellar PC culture; but it turned out to be very challenging. Meanwhile we have good experience in culturing cortical neurons. Although our in vitro data alone could not differentiate whether Sp1-STMNs is operational in PC development or degeneration, together with in vivo data (e.g., Figure 4C-D) and the discussion of literatures, we can comfortably conclude that proper Sp1-STMNs very likely contribute to neuroprotection. Nevertheless, we have carefully rephrased some sentences to reflect certain limitations (Discussion).

6) Please clarify how the ChIP datasets in Figure 4E-F are compared between groups. Why is there no variability in the control group? For ChIP-qPCR data, it would be preferable to show the results as % of input, rather than binding enrichment. Additionally, while the sequences for ChIP-qPCR primers are provided, the authors should clarify where these primers target within the genome, and whether site contains a known SP1 motif.

We now normalized our ChIP-qPCR as % of Input (Figure 4F-G). Per request, we have now also indicated the position where the primers are located in the promoter proximity of STMN3 and 4 in Figure 4—figure supplement 1G. Indeed, these primers’ sequence contain the Sp1 consensus (see Figure 4—figure supplement 1G).

7) The data showing siRNA-mediated Trrap knockdown (Figure 5—figure supplement 1A) is not convincing – summary data and statistical comparisons should be shown for manipulations as performed in the in vitro experiments. Similarly, it is not clear how transfection experiments would lead to robust knockdown. Typically, neuronal transfection experiments suffer from a very low efficiency, meaning that manipulations would only occur in a very small fraction of cells. If this is not the case here, data for transfection efficiency should be provided.

We repeated the siRNA knockdown and provided quantification with statistical analyses in Figure 5—figure supplement 1A. In the rescue experiment, we transfected the siRNA constructs together with a GFP-expressing vector. Only GFP+ cells were used for analyses. We have clarified this point in the figure legend.

8) The Title of the manuscript should be revised to meet eLife guidelines. Specifically, the Title should avoid use of dashes, acronyms, or unfamiliar abbreviations (unless needed for scientific reasons). Please revise your Title with this advice in mind. Additionally, the Title and/or Abstract should provide a clear indication of the biological system under investigation (i.e., species name or broader taxonomic group, if appropriate). Please revise your Title and/or Abstract with this advice in mind.

We followed the advice/guideline and modified the Title and Abstract.

[Editors' note: further revisions were suggested prior to acceptance, as described below.]

Essential revisions:1) Replace all figures with higher resolution images – the current versions appear pixelated upon zooming in to see details. Vector images are preferred where possible.

The original quality of images seems to be fine. The quality problem might be due to the problem of compressed images during conversion of uploaded files. We have readjusted (vector images) where possible and uploaded all figures.

2) In relation to comment 4: Conclusions regarding neurodegeneration are still too strong and the answer provided in response to the reviewers is not fully reflected in the Discussion of the revised manuscript.

We showed clearly a progress neurodegeneration in Trrap deleted Purkinje cell mouse model (Trrap-PCD). However, due to technical limitation, we could not fully prove the molecular mechanism (SP1-STMNs) on the Trrap-PCD model, although we verified STMNs downregulation in their cerebellum. We agree some of the statements are too strong. Following Editors’ suggestion, we have deleted some repetitive conclusions and tried to soften the statement in the relevant Discussion parts.

3) The authors still do not address the lack of variability in control groups in several figures, including:-Figure 4B – from blot images in Figure 4A, there seems to be sample variability in controls, but the individual data points look exactly the same. The same applies to Figure 5—figure supplement 1A.

We have explained the quantification of these figures. We modified presentation of these figures to reflect the data accordingly (Figure 4B, 4H and Figure 5-figure supplement 1A).

-Individual data points are shown for some graphs, but not for others. Individual data points should be included for all graphs.

Following the Editors’ suggestion, we include data points all bar graphs wherever possible in the revised figures.